# Representation of features as images with neighborhood dependencies for compatibility with convolutional neural networks

Omid Bazgir[1], Ruibo Zhang[1], Saugato Rahman Dhruba[1], Raziur Rahman[1], Souparno Ghosh[2,3] & Ranadip Pal [1✉]

Deep learning with Convolutional Neural Networks has shown great promise in image-based classification and enhancement but is often unsuitable for predictive modeling using features without spatial correlations. We present a feature representation approach termed REFINED (REpresentation of Features as Images with NEighborhood Dependencies) to arrange high-dimensional vectors in a compact image form conducible for CNN-based deep learning. We consider the similarities between features to generate a concise feature map in the form of a two-dimensional image by minimizing the pairwise distance values following a Bayesian Metric Multidimensional Scaling Approach. We hypothesize that this approach enables embedded feature extraction and, integrated with CNN-based deep learning, can boost the predictive accuracy. We illustrate the superior predictive capabilities of the proposed framework as compared to state-of-the-art methodologies in drug sensitivity prediction scenarios using synthetic datasets, drug chemical descriptors as predictors from NCI60, and both transcriptomic information and drug descriptors as predictors from GDSC.

[1] Department of Electrical and Computer Engineering, Texas Tech University, 1012 Boston Ave, Lubbock, TX 79409, USA. [2] Department of Mathematics and Statistics, Texas Tech University, 1108 Memorial Circle, Lubbock, TX 79409, USA. [3] Department of Statistics, University of Nebraska-Lincoln, 3310 Holdrege St, Lincoln, NE 68503, USA. ✉email: ranadip.pal@ttu.edu

I n recent years, machine learning (ML) has produced numerous insights from the surge of data generated in diverse areas. For instance, bioinformatics and computational biology have benefited from the availability of high-throughput information from different genomic characterization levels, such as genome, transcriptome, proteome, and metabolome. These large datasets often have the issue of numerous features with limited samples that necessitates the use of feature selection prior to modeling. A predictive modeling framework with high predictive accuracy and inbuilt feature engineering mechanism can be highly useful in such circumstances. In large pharmacogenomic studies, in order to predict drug efficacy based on genomic characterizations, various ML approaches, such as elastic net (EN), random forest (RF), support vector machine (SVM), kernelized Bayesian multitask learning, and so on have been proposed[1–4], where an initial feature selection/extraction step is paramount before model building. Although, sparse linear regression approaches, such as LASSO and EN do offer embedded feature selection, but the accuracy of the models are significantly lower than the ensemble, kernel, and nonlinear regression approaches under model misspecification[1]. In contrast, deep convolutional neural network (CNN) has the potential to provide high-accuracy prediction, while automatically discovering multiple levels of joint representation of the data, and thus eliminating the need for feature engineering[5]. CNN bypasses the a priori manual extraction of features by learning them from data[6]. Furthermore, their representational richness often allows capturing nonlinear dependencies at multiple scales[7] and minimizing generalization error rather than the training error[8].

CNN-based deep learning methods have shown improved performance in various applications, such as speech recognition, object recognition[9], natural language processing[10], genomics, and cancer therapy[11,12]. Deep (multilayered) neural networks are especially well-suited for learning representations of data that are hierarchical in nature, such as images or videos[13]. CNN-based methods have achieved close to human-level performance in object classification, where a CNN learns to classify the object contained in an image[14]. In the field of computational biology, Alipanahi et al.[11] used a 1D CNN architecture to predict specificities of DNA and RNA-binding proteins by directly training on the raw sequence data. Note that 1D CNN can be directly applied to scenarios where the features have relationships with neighbors like DNA or RNA sequences. However, a 1D CNN will not be highly effective in scenarios where ordering of features does not describe the dependencies among features. For instance, gene expression values for biological samples or molecular descriptors for chemical compounds, in their raw form, do not exhibit any form of ordering, and therefore, not amenable to a 1D CNN.

On the other hand, if the predictors are presented in 2D form, i.e., images, a CNN is often effective due to leveraging the spatial correlation among the neighbors to reduce the number of model parameters compared to a fully connected network, by utilizing the convolution operation and parameter sharing. In classification setup, Coudray et al.[12] demonstrated the efficiency of this approach to distinguish the most prevalent lung tumor subtype from normal lung tissues by using whole slide images from The Cancer Genome Atlas and validating on independent histopathology images. Thus, the ability to represent a collection of potentially high-dimensional scalar features as images, with correlated neighborhoods, has the potential of benefiting from the automated feature extraction and high-accuracy prediction of CNN-based deep learning. To our knowledge, the only other approach for representing data as images is OmicsMapNet[15] that was proposed at the same time we were developing our REFINED idea. OmicsMapNet uses TreeMap[16] to rearrange omics data into 2D images that requires preliminary knowledge extracted from Kyoto Encyclopedia of Genes and Genomes, and therefore, cannot be used in absence of ontology knowledge, or in cases of non-omics features such as drug descriptors[17–19].

In this paper, we present a methodology termed REFINED (REpresentation of Features as Images with NEighborhood Dependencies), for representing high-dimensional feature vectors as mathematically justifiable 2D images that can be processed by standard CNN-based deep learning methodologies. We illustrate the advantages of our proposed framework in terms of accuracy and Bias characteristics on both synthetic and pharmacological datasets from public repositories, such as NCI60 Human Cancer Cell Lines Screen[20] and Genomics of Drug Sensitivity in Cancer (GDSC)[21].

## Results

In this section, we report the performance of our REFINED-CNN methodology on the three different scenarios using a synthetic dataset, NCI60, and GDSC. We compare these performances with the performances for standard ML approaches, such as EN, logistic regression (LR), RF, SVM classifier or as regressor (SVR), and deep artificial neural network (ANN). The detailed description of the datasets along with the model specifications and architectures are provided in the "Methods" section. Moreover, for NCI60 and GDSC scenarios, we also compare REFINED-CNN performances with two other CNN-based approaches—Random-CNN and principal component analysis (PCA)-CNN—with input images being generated by random permutation and PCA on the features, respectively, as elaborated below in the "Alternative image generation approaches" section.

**Evaluation metrics**. We used two sets of performance evaluation metrics to asses the results for scenarios in two modeling paradigms—regression and classification.

*Metrics for regression*: We used normalized root mean square error (NRMSE), and Pearson correlation coefficient (PCC) between the predicted and observed values along with the reduction in Bias in prediction for evaluation. NRMSE is the ratio of the root mean square error (RMSE) for a given model over the error for the observed mean as prediction, and gives an estimate of the capabilities of the given model in minimizing the generalization error normalized to the variance in the observed response. PCC gives the degree of collinearity between the observed and predicted responses, and a high value is desired since a lower PCC value implies a lack of collinearity often indicative of model misspecification and poor predictive capability. We represent the Bias in prediction as the slope of the best-fitted line through the prediction residuals, where $x$-axis represents the observed response, or in other words, the tangent of the bias angle, $\theta$ in degrees, between the best-fitted line and the observed values. This ensures that the bias is in [0, 1] where a smaller angle is desired, since an unbiased model is expected to produce $\theta_{unbiased} = 0°$ or bias = 0. We also used the normalized mean absolute error (NMAE) metric for a few comparisons that is defined similiarly to NRMSE except for using MAE instead of RMSE. Representing the observation, prediction and residual vectors as $\mathbf{y}$, $\hat{\mathbf{y}}$, and $\boldsymbol{\epsilon}$, the corresponding mean values as $\bar{y}$, $\bar{\hat{y}}$, and $\| \cdot \|, | \cdot |$ as Euclidean norm and absolute value operations, these metrics are defined as

$$\text{NRMSE} = \frac{\text{RMSE for given model}}{\text{RMSE for intercept-only model}} = \frac{\| \mathbf{y} - \hat{\mathbf{y}} \|}{\| \mathbf{y} - \bar{y} \|}, \quad (1)$$

$$\text{PCC} = \frac{\text{cov}(\mathbf{y}, \hat{\mathbf{y}})}{\sqrt{\text{var}(\mathbf{y}) \, \text{var}(\hat{\mathbf{y}})}} = \frac{(\mathbf{y} - \bar{y})^{\text{T}} (\hat{\mathbf{y}} - \bar{\hat{y}})}{\| \mathbf{y} - \bar{y} \| \| \hat{\mathbf{y}} - \bar{\hat{y}} \|}, \quad (2)$$

$$\text{Bias} = \tan\theta; \quad \theta = \angle(\mathbf{y}, \boldsymbol{\epsilon}) = \angle(\mathbf{y}, \mathbf{y} - \hat{\mathbf{y}})$$

$$= \cos^{-1}\left(\frac{\mathbf{y}^T(\mathbf{y} - \hat{\mathbf{y}})}{\|\mathbf{y}\|\|\mathbf{y} - \hat{\mathbf{y}}\|}\right), \tag{3}$$

$$\text{NMAE} = \frac{\text{MAE for given model}}{\text{MAE for intercept-only model}} = \frac{|\mathbf{y} - \hat{\mathbf{y}}|}{|\mathbf{y} - \bar{y}|}. \tag{4}$$

*Metrics for classification*: The discriminative models for the sensitive and resistant drugs in NCI60 were evaluated using accuracy, precision, recall, $F_1$-score, and the area under the receiver operating characteristics (AUROC) metrics. Accuracy is defined as the fraction of correctly identified sensitive (true positive) and resistant (true negative) drugs in the total set of drugs. Precision is the model capability of predicting the positive instances, i.e., the fraction of correctly identified sensitive drugs in all drugs identified as sensitive. Recall or true positive rate (TPR) is the model sensitivity to identifying the positive instances, i.e., the ratio of two numbers—correctly identified sensitive drugs and actual sensitive drugs. $F_1$-score is a measure of binary classification accuracy and defined as the harmonic mean of the precision and recall of the classifier. Finally, AUROC is the highest AUROC curve in a plot containing multiple ROC curves, and thus, multiple potential candidate models. Note that, a ROC curve displays the model TPR and FPR (false positive rate; the fraction of false positives in all positive identification, i.e., the fraction of sensitive drugs misidentified as resistant in all drugs identified as sensitive) values at various threshold levels. Representing the total count of true positive, true negative, false positive, and false negative (sensitive drugs misidentified as resistant) in prediction as TP, TN, FP, and FN, these metrics are defined as

$$\text{Accuracy} = \frac{\text{TP} + \text{TN}}{\text{TP} + \text{TN} + \text{FP} + \text{FN}}, \tag{5}$$

$$\text{Precision} = \frac{\text{TP}}{\text{TP} + \text{FP}}, \tag{6}$$

$$\text{Recall} = \frac{\text{TP}}{\text{TP} + \text{FN}}, \tag{7}$$

$$F_1\text{-score} = \left(\text{Precision}^{-1} + \text{Recall}^{-1}\right)^{-1} = \frac{2\text{TP}}{2\text{TP} + \text{FP} + \text{FN}}, \tag{8}$$

$$\text{FPR} = \frac{\text{FP}}{\text{FP} + \text{TN}}. \tag{9}$$

*Robustness metrics*: Besides these paradigm-specific metrics for each task, we also calculated the 95% confidence interval (95% CI) for each individual metric to asses the prediction robustness. For regression tasks, we applied a pseudo *jackknife-after-bootstrap*[3,22] approach, where multiple bootstrap sets are selected from the complete set of test samples and a corresponding error metric distribution is used to calculate the 95% CI for a given sample (i.e., a cell line in NCI60 or a cell line–drug combination in GDSC). For classification tasks, we used the *binomial proportion confidence interval*[23,24].

Furthermore, for regression tasks, we used Gap statistics[25] to report the significance of the difference in performance across methods. We paired each individual model with a null model[1] through bootstrap sampling on the dose–response values of the test set along with the corresponding predicted values. The null model randomly predicted dose–response values for test set using the distribution of the training set dose–responses, and this process is repeated for 10,000 times to generate individual pairs of distributions for NRMSE, PCC, and bias metrics for each regression model–null model pair. Finally, we used $k$-means clustering ($k = 2$) to generate two cluster centroids for each pair of distributions and the difference between these centroids represents the performance difference. In addition, all models were subjected to a robustness analysis[1] through calculating the number of times REFINED-CNN outperformed the competing models in those 10,000 repetitions.

**Alternative image generation approaches**. For comparison purposes, we consider two alternative image generation methods based on (i) random projections and (ii) PCA representation. In the random projection approach, we assume each image is a matrix and the location of each entry in the vector is randomly mapped to a location in the matrix. Hence, we placed each element of the drug descriptors or gene expression values on the image coordinates one after another. In the PCA-based approach, we employed PCA[26] that is used extensively for data visualization or dimensionality reduction (DR). In PCA, each sample could be represented on a 2D plane aligned with the first two major eigenvectors (or principal components; PCs) of the data covariance matrix. We used the sets of first two PCs from the covariance matrices of our datasets, as feature coordinate sets for PCA-based image generation. Some example images from the random projection and PCA-based procedures are shown in Fig. 1, which afterward, are used to train individual CNNs, dubbing the complete methodologies as Random-CNN and PCA-CNN, respectively.

**Synthetic data**. We offer comparative performances of the candidate models on a simulated dataset. First, we generate a lexicographic ordering of $P$ features. The features are generated from a zero-mean Gaussian process with covariance depending only on the lexicographic distance between the features. In each case, a subset of features were randomly selected as spurious. Subsequently, random weights generated fo non-spurious features were used to generate the target values. The generated target values were normalized in [0, 1]. We have used REFINED to generate images for different $N$ and/or $P$ scenarios and then trained a CNN for each. In each case, the same dataset was used to train RF, SVR, and ANN for comparison, and performances were evaluated via fivefold cross-validation. The results are summarized in Fig. 2 as heatmaps, where the green regions represent the cases where the REFINED-CNN NRMSE is less than the competing models. The separate heatmaps for all model results are provided in Supplementary Figs. 1–4. The heatmaps clearly show that the REFINED-CNN methodology outperforms others when the feature and sample sizes are relatively large ($P > 100$, $N > 600$), regardless of the amount of spurious features present in the dataset. Furthermore, we observe that the performance of the posited methodology improves as $N$ increases. Chen et al.[27] also reported that the performance of their unidimensional scaling (UDS)-based projection improved with increase in both $N$ and $P$, and therefore, our findings suggest that our second-order REFINED approximation is in agreement with their first-order stringing approximation[27]. Moreover, as the ratio of spurious features increases, we observe REFINED-CNN to increasingly outperform the competing models even without any feature preselection step. This exercise demonstrates the ability of our approach to automatically remove spurious features without performing an explicit feature selection a priori. Also, by comparing the REFINED-CNN vs. ANN heatmaps, we observe that increasing the sample size reduces the gap between their performance, as more samples are available to train the large number of ANN parameters.

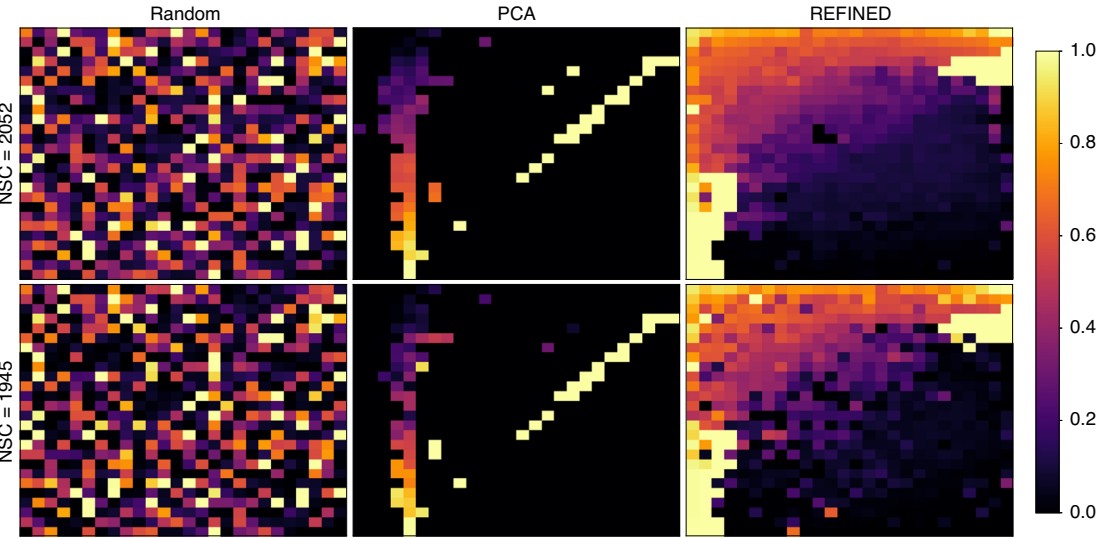

**Fig. 1 Examples of image generation through different approaches.** Illustration of some example images for NCI60 drug descriptors, where each title denotes the associated image generation method, and each drug is denoted by a unique NSC ID[54] assigned by the Development Therapeutic Program (DTP) of National Cancer Institute (NCI) to a chemical agent or product ranging from small molecules to compounds.

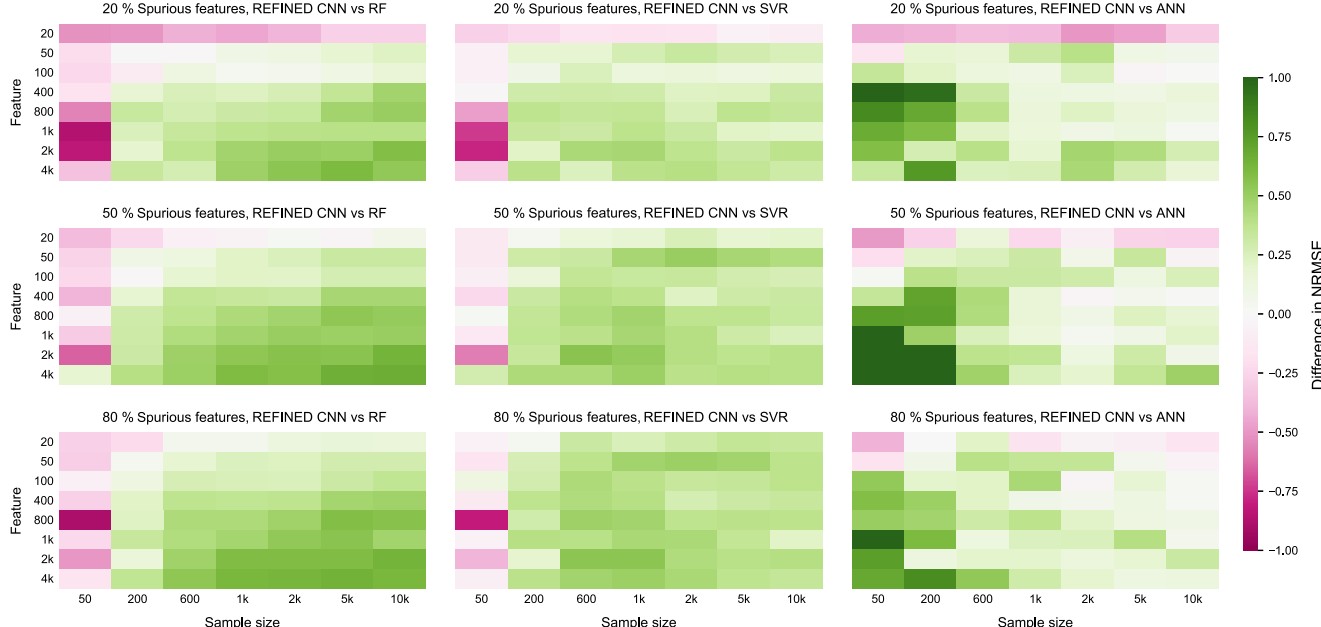

**Fig. 2 Performance comparison for varying sample and feature sizes.** Differences in NRMSE values for REFINED-CNN with three competing models (RF, SVR, and ANN) for different sample sizes and feature sizes containing varying degrees of spuriosity. Each green heatmap region denotes a case where REFINED-CNN outperforms the other models.

We next investigated the effect of the REFINED-CNN approach on the bias characteristics of the prediction. Supplementary Fig. 5 shows the scatter plot of prediction vs. observed responses for the four models when 80 and 20% of the features are spurious. Clearly, the scatter plot for REFINED-CNN closely follows a straight line with unit slope indicating the predictive accuracy of our approach. RF and SVR reveal their well-known tendency to underpredict the higher values and overpredict the lower values in observation[28]. REFINED-CNN bias is also lower than the bias observed for the ANN scenario. Thus, it appears that REFINED-CNN approach can automatically improve the bias in prediction that some of the existing models are known to suffer from.

**NCI60 classification tasks.** We investigated the discriminative power of REFINED-CNN approach to identify the resistant and sensitive (nonresistant) drugs on different NCI60 cell lines, as compared to the other models. The threshold for defining resistant and sensitive classes was selected based on the drug sensitivity distribution shown in Supplementary Fig. 6. A drug is considered resistant if the corresponding normalized $GI_{50}$ (NLOGGI50) values are <4.25 and sensitive otherwise. Since each unique drug for a particular cell line is a sample in this scenario, and we have sufficiently large number of drugs for each cell line. We randomly considered 80% of the drugs for training, 10% for validation, and 10% for testing. As shown in the Fig. 3a, the REFINED-CNN outperforms other classifiers for all 17 cell lines. The mean

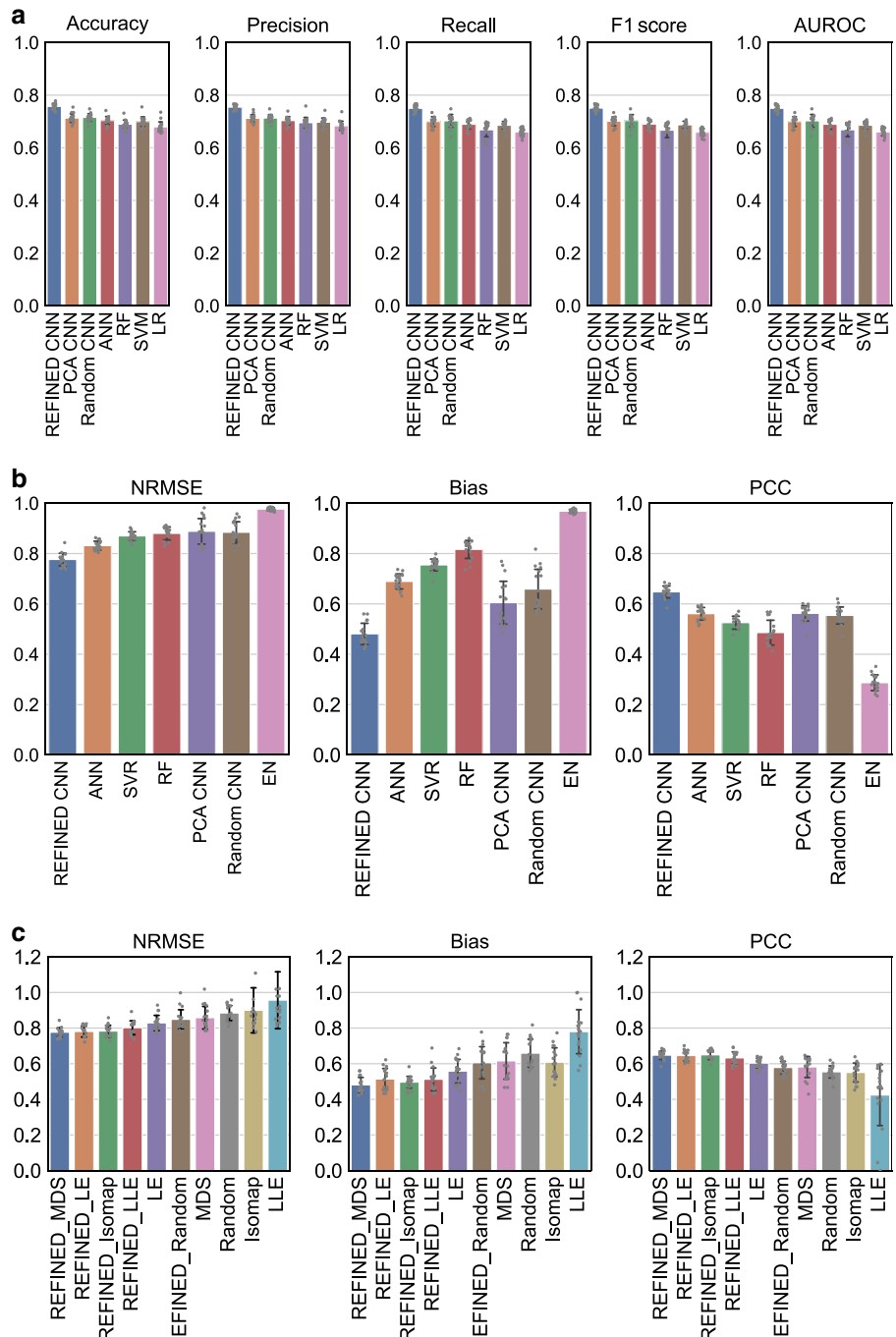

**Fig. 3 Comparative performance of REFINED-CNN for the NCI60 dataset.** The predictive model associated with each bar is defined under it. **a** Summary of REFINED-CNN classifier performance compared to six other classifiers for all 17 cell lines. **b** Summary of REFINED-CNN performance compared to six other regression models for all 17 cell lines. **c** Summary of REFINED ablation study results on the regression task where REFINED_MDS represents our proposed approach.

classification accuracy of REFINED-CNN, $ACC_{REFINED-CNN} =$ 75.4% is considerably higher than the mean accuracy obtained for the other six models: $ACC_{Random-CNN} = 71.6\%$, $ACC_{PCA-CNN} =$ 71.7%, $ACC_{ANN} = 70.3\%$, $ACC_{RF} = 70.0\%$, $ACC_{SVM} = 69.0\%$, and $ACC_{LR} = 67.9\%$. For each model, we also report the precision, recall, $F_1$-score, and AUROC values, all of which demonstrate the superior discriminative capabilities of REFINED-CNN. The detailed classification results along with 95% CI values are provided in Supplementary Tables 1–2.

To compare the statistical significance of the difference in performance of the competing classifiers with REFINED-CNN, we used McNemar's test, a paired nonparametric statistical hypothesis test, that evaluates whether two models disagree in the same way or not[29]. We performed a pairwise comparison of each classifier with REFINED-CNN by forming a contingency table and using the null hypothesis that both models disagree by the same amount with a $p$-value cutoff of 0.05. The complete results for these comparisons are provided in Supplementary Table 3.

**Table 1 Effect of bootstrap data augmentation on REFINED-CNN performance.**

| Cell line | Sample size (O) | Sample size (B) | NRMSE (O) | NRMSE (B) | NMAE (O) | NMAE (B) |
|---|---|---|---|---|---|---|
| SNB_78 | 13940 | 19613 | 0.784 | 0.744 | 0.713 | 0.688 |
| MDA_MB_435 | 36868 | 59570 | 0.787 | 0.762 | 0.739 | 0.729 |
| NCI_ADR_RES | 37156 | 59250 | 0.798 | 0.755 | 0.745 | 0.717 |
| 786_0 | 49344 | 76908 | 0.752 | 0.713 | 0.688 | 0.663 |
| COLO_205 | 48946 | 75158 | 0.741 | 0.722 | 0.664 | 0.660 |

*O original dataset, B bootstrap-augmented dataset.*

We observe that the REFINED-CNN classifier performance is significantly different from the other six classifiers—LR, RF, SVM, ANN, Random-CNN, and PCA-CNN.

**NCI60 regression tasks**. The NCI60 dataset was randomly partitioned into 80, 10, and 10% segments for training, validation, and test, and the same sets were used for model comparison. The performance of each model was evaluated using NRMSE, PCC, and bias. Figure 3b provides a bar plot summary of the performance of each model (the detailed performance is provided in Supplementary Table 4 along with the corresponding 95% CI values in Supplementary Fig. 7). We note that CNN again outperforms all the competing models in all 17 cell lines with an average improvement of 6–20% in NRMSE, 8–36% in PCC, and 12–38% in bias. To explore the performance variations with different random projections, we tried different projections and the corresponding results in Supplementary Table 5 bolster the notion that there exists no random feature map that can potentially outperform REFINED as feature representation.

We used Gap statistics and robustness analysis as described in the "Evaluation metrics" section to compare the REFINED-CNN prediction with other competing models. The robustness analysis results in Supplementary Table 6a indicate that REFINED-CNN has better performance in terms of NRMSE for 89.5–100% of the times, PCC for 94.4–100% of the times, and bias for 91.1–100% of the times on average. The Gap statistics results in Supplementary Table 6b strengthens this conclusion for both average and per cell line scenarios. The NRMSE and PCC distributions for all seven models along with the null models are displayed for three NCI60 cell lines in Supplementary Fig. 8.

**Data augmentation**. We analyzed the effect of data augmentation on the performance of REFINED-CNN, using samples from the less represented regions in the sensitivity distribution displayed in Supplementary Fig. 6. The massive point mass associated with the nonsensitive (resistant) drugs severely impacts a global regression model for modeling NLOGGI50. This problem is analogous to the zero-inflation problem in classical statistical literature, where the discrete point mass is modeled separately from the continuous part (see refs. [30,31] and references therein). In our situation, it boils down to a classification into sensitive/resistant category followed by a regression for the sensitive category. We have already demonstrated superiority of REFINED-CNN in both scenarios in Supplementary Tables 1 and 4. Now, we explore if the REFINED-CNN performances could be improved by oversampling the sensitive category to arrive at a more balanced dataset. To that end, we used a version of Synthetic Minority Oversampling Technique (SMOTE)[32] to generate bootstrap replicates from the sensitive category. The NRMSE and NMAE improvements of REFINED-CNN on five different cell lines are illustrated in Table 1. The bootstrap data augmentation systematically decreases the NRMSE and NMAE for the cell lines, indicating the negative impact of the point mass in the response distribution.

**Sample size analysis**. Deep CNN models are expected to perform better with larger number of samples, as compared to smaller number of samples. Therefore, we trained our model on different portions of training sets for randomly selected cell lines to test this hypothesis. We trained our model on 20, 40, 60, and 80% of the available drugs applied on the selected cell lines, and kept the rest of the data for testing, considering NRMSE as a comparison metric. The results of five cell lines are summarized in Supplementary Fig. 9 and Supplementary Table 7 that illustrates that REFINED-CNN outperforms the other models as sample size increases. This trend was also observed for the synthetic data in Fig. 2.

**Model stacking analysis**. To explore whether stacking of multiple models can improve prediction performance, we used linear stacking to combine various models and predict the test set sensitivity values, while using the validation set predictions to calculate stacking weights for each model. We used three different combinations of models—(i) non-CNN stacking using RF, SVR, ANN, and EN, (ii) CNN-based stacking using PCA-CNN, Random-CNN, and REFINED-CNN, and (iii) all model stacking. Supplementary Fig. 10 shows the performances of these three stacked models with the following mean NRMSE values: $NRMSE_{all} = 0.738$, $NRMSE_{CNN} = 0.744$, and $NRMSE_{nonCNN} = 0.837$. Comparing these results with the mean NRMSE of each model in Supplementary Fig. 10 reveals a significant improvement in performance for stacked models compared to the individual non-CNN models. Note that, the mean NRMSE for REFINED-CNN in Fig. 3b is also significantly lower than the NRMSE for the non-CNN stacked model.

**Bias analysis**. We reinvestigate the effect of the REFINED-CNN approach on the bias characteristics of the prediction using the NCI60 dataset. Supplementary Fig. 11 displays the residual vs. observed plots for all seven models. Similar to the synthetic data scenario described above, Supplementary Fig. 11: first row shows that REFINED-CNN has the lowest bias (i.e., the smallest bias angle): $\theta_{REFINED-CNN} = 20.2°$ compared to the rest: $\theta_{Random-CNN} = 32.8°$, $\theta_{PCA-CNN} = 23.8°$, $\theta_{ANN} = 29.2°$, $\theta_{RF} = 34.6°$, $\theta_{SVR} = 33.5°$, and $\theta_{EN} = 43.1°$. To investigate whether bias correction erodes the advantage of REFINED-CNN in terms of bias, we considered the *BC1 bias correction* algorithm proposed by Zhang et al.[33], where we fit a linear regression model on the residuals. The results shown in Supplementary Fig. 11: second row illustrates the superiority of REFINED-CNN in terms of bias reduction even after additional bias correction is applied to the competing models.

**GDSC sensitivity prediction**. We consider the application of REFINED-CNN in integrating of two types of heterogeneous datasets. Our predictors consist of (i) PaDEL descriptors[34] representing the chemical compounds, and (ii) gene expression representing genetic characterization profiles. The response consists of the $IC_{50}$ values for a particular drug–cell line combination. We used the REFINED approach to generate the images

**Table 2 REFINED-CNN performance comparison with competing models for the GDSC dataset.**

| Model | NRMSE (20%) | NRMSE (50%) | NRMSE (80%) | PCC (20%) | PCC (50%) | PCC (80%) | Bias (20%) | Bias (50%) | Bias (80%) |
|---|---|---|---|---|---|---|---|---|---|
| EN | 0.890 | 0.889 | 0.887 | 0.488 | 0.484 | 0.486 | 0.848 | 0.849 | 0.840 |
| RF | 0.609 | 0.620 | 0.569 | 0.797 | 0.785 | 0.821 | 0.433 | 0.417 | 0.337 |
| SVR | 0.750 | 0.742 | 0.525 | **0.847** | 0.845 | 0.853 | 0.257 | 0.273 | 0.241 |
| ANN | 1.407 | 0.475 | 0.435 | 0.519 | 0.883 | 0.901 | 0.784 | **0.153** | 0.233 |
| Random-CNN | 0.579 | 0.456 | 0.441 | 0.836 | 0.892 | 0.903 | 0.215 | 0.193 | 0.222 |
| PCA-CNN | 0.612 | 0.461 | 0.443 | 0.820 | 0.891 | 0.901 | **0.201** | 0.228 | **0.179** |
| REFINED-CNN | **0.541** | **0.439** | **0.414** | 0.845 | **0.899** | **0.911** | 0.255 | 0.173 | 0.197 |

The numbers in parentheses indicate the percentages of the available data used for training.
*EN* elastic net, *RF* random forest, *SVR* support vector regression, *ANN* artificial neural networks, *Random-CNN* random mapping based convolutional neural network, *PCA-CNN* principal component analysis based convolutional neural networks, *REFINED-CNN* proposed REFINED approach-based convolutional neural networks. Bold values indicate the best performances.

corresponding to the gene expressions for each cell line and drug descriptors for each drug compound in the GDSC dataset.

Considering 222 drugs and around 972 cell lines for each drug, the total number of samples in the dataset is close to 177 K. We randomly divided the dataset into 80% training, 10% validation, and 10% test sets, where each set covariates contains 1211 genes and 992 chemical drug descriptors. Supplementary Fig. 12a represents the scatter plot of the $\log IC_{50}$ prediction (top row) using REFINED-CNN, Random-CNN, PCA-CNN, ANN, RF, SVR, and EN along with the corresponding residual plots (bottom row). Table 2 provides a performance summary for each model using NRMSE, PCC, and bias. Similar to NCI60, we used Gap statistics and robustness analysis to further compare the models in Supplementary Table 8. The Gap statistics distribution plots per metric per model-pair with corresponding cluster centroids are provided in Supplementary Figs. 13–15.

As shown in Table 2, REFINED-CNN model achieves improvement compared to other models in the range of 3–47% for NRMSE, and 1–42% for PCC. We also train REFINED-CNN on 50 and 20% of the available samples, and compared the performance with the competing models. Since the CNN architecture (see Supplementary Fig. 16) trained on the 80% REFINED images was too complex to be trained on the 20% REFINED images, we sought to reduce the network complexity for both REFINED-CNN and ANN models by removing certain layers (namely the last convolution layer and the following dense layer for CNN and the first two dense layers for ANN). The detailed results are provided in Supplementary Table 9, while the corresponding prediction and residual scatter plots are in Supplementary Fig. 12b, c.

We also incorporated a larger feature size (i.e., number of genes) to assess the effect of input image size. We selected feature subsets of size, $P = 2147$, 2985, 4271, 8048 ($\approx$2000, 3000, 4000, 8000) and created corresponding REFINED images to train the same CNN architecture that we used with 1211 genes, and the other competing models with same hyperparameter sets. However, the SVR model crashed after several hours due to large memory requirement during kernel estimation via pairwise distance computation, and therefore, we resorted to use a bag of SVRs with selected subsets of 400 features (appended gene expression and drug descriptors) to train 100 models in parallel. The complete results are provided in Supplementary Fig. 17 and Supplementary Table 9 along with the 95% CI values for 80% REFINED images in Supplementary Fig. 18. We observed that the trend of REFINED-CNN outperforming other models is maintained irrespective of the training feature size.

**Comparison with state-of-the-art**. We have compared REFINED-CNN with two state-of-the-art approaches. The first

**Table 3 Comparison of REFINED-CNN with DRF and HGNN for GDSC dataset.**

| Model | NRMSE | PCC | Bias |
|---|---|---|---|
| REFINED-CNN | **0.414** | **0.911** | **0.197** |
| DRF | 0.986 | 0.169 | 0.976 |
| HGNN | 0.637 | 0.805 | 0.446 |

*REFINED-CNN* proposed REFINED approach-based convolutional neural networks, *DRF* Deep-Resp-Forest, *HGNN* heterogeneous graph neural networks. Bold values indicate the best performances.

one is the Deep-Resp-Fores (DRF) proposed by Su et al.[35]. DRF is a deep cascaded forest designed to classify anticancer drug response as sensitive or resistant, using heterogeneous input sets of gene expression and copy number alteration (CNA) from GDSC. We switched the RF classifier in DRF to a RF regressor for our application and used drug descriptors in place of CNA for training. The second approach is the heterogeneous graph neural networks (HGNN) proposed by Lim et al.[36]. HGNN automates the feature engineering task by aggregating feature information of neighborhood nodes, where the input data of the network are from different sources[37], and embeds the 3D structural information of protein–ligand complexes in distance matrix to predict drug–target interactions. We trained their network by encompassing gene–drug information in distance matrix to predict drug sensitivity for GDSC. The comparative performance for DRF and HGNN with REFINED-CNN is provided in Table 3 along with the corresponding robustness analyses in Supplementary Table 8. As the results indicate, REFINED-CNN outperforms both DRF and HGNN for GDSC drug sensitivity prediction.

**Ablation study**. The main components of REFINED are a DR followed by a search optimization algorithm. We used Bayesian multidimensional scaling (BMDS) as a global distortion minimizer and hill climbing as local adjustment to reach a locally optimal configuration among multiple automorphs. To investigate the contribution of each component to REFINED, we have evaluated the effect of using different approaches in each step. For the first step, we replaced BMDS with different DR techniques, such as isomap[38], linear local embedding (LLE)[39], and Laplacian eigenmaps (LE)[40]. Isomap generalizes multidimensional scaling (MDS) with using geodesic distance (GD) in nonlinear manifolds rather than Euclidean distance (ED), where GD is approximated as a sum of ED values. LLE is a local technique that tries to reconstruct each sample based on the k-nearest samples in a lower dimension locally. LE is similar to LLE except it uses *Laplacian graph* to reconstruct the k-nearest neighbors, where k eigenvectors corresponding to the k smallest eigenvalues are

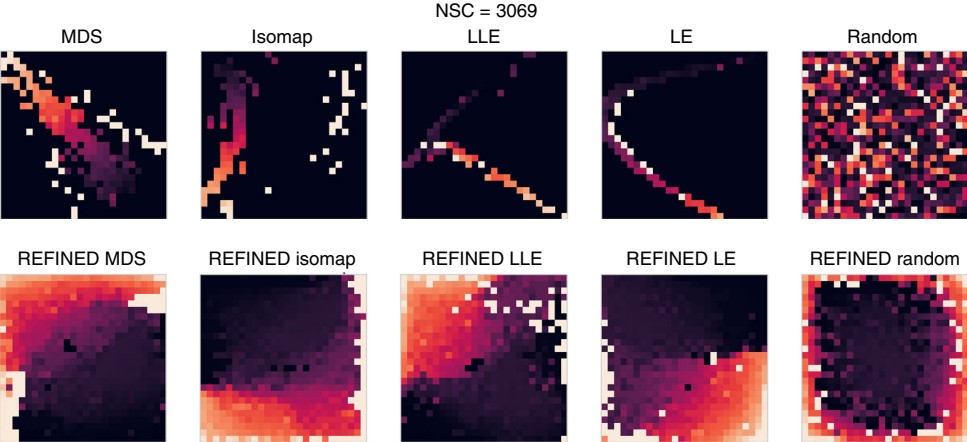

**Fig. 4 Example of REFINED images created through different 2D mappings.** Illustration of images generated for an example drug from NCI60 using the REFINED approach initialized with different dimensionality reduction techniques, and before (top row) and after (bottom row) applying hill climbing to eliminate feature overlapping.

**Table 4 Ablation study for REFINED using regression on NCI60 dataset.**

| Mapping | NRMSE (N) | NRMSE (HC) | PCC (N) | PCC (HC) | Bias (N) | Bias (HC) |
|---|---|---|---|---|---|---|
| BMDS | 0.858 | **0.776** | 0.582 | 0.647 | 0.616 | **0.481** |
| Isomap | 0.900 | 0.783 | 0.551 | **0.649** | 0.608 | 0.496 |
| LLE | 0.956 | 0.802 | 0.424 | 0.632 | 0.780 | 0.512 |
| LE | **0.829** | 0.781 | **0.603** | 0.645 | **0.558** | 0.514 |
| Random | 0.884 | 0.849 | 0.554 | 0.578 | 0.659 | 0.605 |

*N* no search optimization, *HC* hill climbing, *BMDS* Bayesian multidimensional scaling, *LLE* linear local embedding, *LE* Laplacian eigenmaps. Bold values indicate the best performances.

preserved for embedding[41]. To investigate the contribution of the search optimization, we initialized each feature location randomly and applied the hill climbing methodology with the objective of minimizing the difference between feature distance matrix (ED) in the initial domain and the created REFINED image. We generated images using all the abovementioned methods and then trained CNNs with the architecture used for NCI60 regression task on the same cell lines. Figure 4 shows the images created using each described method for an example drug from NCI60, while Supplementary Figs. 19 and 20 provides the images for five more drugs.

The mean performances are presented in Table 4 and as bar plots in Fig. 3c. The results demonstrate that REFINED initialized with BMDS provides the smallest NRMSE values compared to the other DR approaches. We also observed that both global and local embeddings do better than random initialization followed by hill climbing. Furthermore, the idea of nonoverlapping pixel locations for the features borne out of the hill climbing step improves the performance of the individual DR approaches. The detailed results are provided in Supplementary Table 10.

### Discussion
This paper presents an approach of converting high-dimensional vectors into images with spatial neighborhood dependency that can be used as input to a traditional CNN architecture. The proposed methodology was conceived from the observation that CNN-based deep learning has increased the prediction accuracy in many scenarios especially when the input are images, but is not usually appropriate when high-dimensional vectors with limited neighborhood correlations are available as input. Our REFINED approach produces a mapping of the input features such that the

spatial neighbors are close by and distant points in the initial feature space are represented by faraway points in the map. Several advantages of the proposed REFINED methodology can be outlined as—the REFINED mapping to a compact image space appears to allow for automated feature extraction using deep CNN architecture. Using a synthetic dataset and varying the amount of spurious features, we observed that REFIEND-CNN is able to significantly outperform other approaches even for scenarios involving larger percentages of spurious features as shown in Fig. 2. The REFINED-CNN methodology provides a gain in predictive accuracy compared to other commonly used approaches, such as ANN, RF, SVM, EN, and LR. We have validated the performance of REFINED-CNN using a synthetic dataset, NCI60 drug response dataset, and GDSC heterogeneous dataset containing molecular descriptors of drugs with transcriptomic expressions of cell lines. REFINED-CNN also outperforms state-of-the-art methods such, as Deep-Resp-Forest[35] and heterogeneous graph networks[36] for the heterogeneous input scenario. REFINED-CNN also outperforms the existing approaches in statistical significance and robustness. REFINED-CNN methodology can also be used to seamlessly combine heterogeneous predictors where each predictor can be mapped to an image, as demonstrated for the GDSC prediction scenario. Perhaps the biggest advantage of REFINED-CNN is that it has the potential to combine multi-type predictors, where some predictors are images, some can be high-dimensional vectors, and others can have functional forms. In principle, each type of predictor can be individually mapped to images and the corresponding images can be used as input in a multi-arm CNN architecture. We observe that REFINED-CNN has better ability to automatically perform bias correction as compared to ANN, RF, and SVR as shown in all three application scenarios (see Supplementary Figs. 5, 11 and 12).

The proposed REFINED approach can also be used for data augmentation by using different realizations of the mapping function as discussed in "Theoretical basis" in the "Methods" section. We have provided a theoretical justification to motivate how the proposed approach can map to an ordering of features if such an ordering exists.

In terms of applications, REFINED can be applied to any predictive modeling scenario where the predictors are high-dimensional vectors without an explicit neighborhood-based dependency structure. We motivated the application scenario through the problem of predicting drug efficacy in cell lines based on genetic characterizations and/or molecular descriptors, where neither input is necessarily ordered based on correlation. Limitations of the approach will include scenarios where the covariance structure of the features is primarily diagonal with limited correlations between variables. Furthermore, the REFINED approach is expected to benefit from the traditional CNN architecture, and thus the performance boost will require a large number of samples typical to a CNN. Also note that, REFINED is a second-order approximation under the Euclidean norm, and therefore, if the predictor space is non-Euclidean, the current form of REFINED will not be suitable.

To summarize, the paper presents an effective tool for feature representation and multi-object regression and classification.

## Methods

In this section, we elaborate on our proposed REFINED algorithm for mapping high-dimensional feature vectors to images, as well as describe the datasets used for performance evaluation followed by the CNN architectures used as the predictive model.

**REFINED**. As mentioned earlier, the main idea of the REFINED-CNN approach is to map high-dimensional vectors to mathematically justifiable images for training by traditional CNN architectures. Evidently, a mapping of features from the high-dimensional vector to a 2D image matrix serially in a row-by-row or column-by-column manner will not guarantee any spatial correlations in the image. Instead, we first obtain the ED matrix of the features and use it as a distance measure to generate a compact 2D feature representation, where neighborhood features are closely related. A potential solution to achieve this 2D projection is to apply DR approach, such as MDS[42] on a distance measure, e.g., ED. However, that will not guarantee that each mapped point will have a unique pixel representation in the image and might result in sparse images due to the overlap[43]. For instance, if we have 900 features, the features can potentially be represented by a $30 \times 30$ matrix and a direct MDS-like approach on a $30 \times 30$ space might not spread out each feature in such a manner that each pixel contains at most one feature. To ensure that the features are spaced out in a discrete grid and to incorporate the discrete nature of the image pixels, we apply a Bayesian version of metric MDS (BMDS). First, we start with the MDS algorithm to create an initial feature map (a 2D space with feature coordinates) that preserves the feature distances in the 2D space with minor computational cost. Next, we apply BMDS to estimate the feature locations on a bounded domain with the constraint that each pixel can at most contain one feature. However, the locations of the features are estimated up to an automorphism, and therefore, we apply a hill climbing algorithm with a cost function that measures the absolute difference in the ED values among the new feature locations (as represented by the 2D image map) to the estimated true distances ($\hat{\boldsymbol{\delta}}$, anticipating the following section) among the features to arrive at an optimal configuration.

More specifically, starting from the BMDS location estimates, we considered all the configurations in the map sequentially in row order. For each feature, we tried different permutations of the features by interchanging the position of the central feature position with its neighboring features, and selected the permutation that minimizes the abovementioned cost function. Once the cost function is minimized, a set of unique coordinates in a 2D space was produced for each feature. Using those coordinates, we mapped the features into a 2D space and create an image per sample. The created images were then used to train a suitable CNN architecture.

The general idea of the REFINED-CNN methodology is shown pictorially in Fig. 5 for the application case of predicting drug efficacy over a cell line, using genetic characteristics of cell lines and molecular descriptors of the drug as predictors. Here, we use the PaDEL software[34] to get the as drug descriptors. In Fig. 5, an example case is shown where $F_{12}$ has been interchanged with its neighboring features, and after each exchange, we checked the similarities (i.e., correlation) among the distances of features from the map and estimated distance matrix of descriptors. If we can find a better exchange case in the feature map, we exchange that feature pair and arrive at a new feature map. The entire process was

repeated iteratively until we reached the optimized feature map that has a distance matrix close to the benchmark distance matrix, $\hat{\boldsymbol{\delta}}$ of the initial features. At the conclusion of this iterative algorithm, we arrive at a REFINED feature map with all features having a unique position in a bounded 2D space, and similar features are placed close by and dissimilar features are placed apart. Without loss of generality, we have considered feature maps on unit square and the BMDS specification induced sparsity in the image.

Figure 1 shows some REFINED images generated for different drugs (represented by distinct NSC IDs). Each image varies from another depending on the values of the PaDEL descriptors of the drug, but the descriptor coordinates are same for all the cases.

**Theoretical basis**. Consider the predictor matrix $\boldsymbol{X} = \{x_{ij}\}$, $i = 1, 2, \cdots, n$; $j = 1, 2, \cdots, p$ with $x_{ij}$ being the value of the $j$th predictor for the $i$th subject. Suppose, the predictors are generated from a latent zero mean, square integrable stochastic process $\{Z(s)\}$ where the index $s$ belongs to a compact subset of $\mathbb{R}^m$. Let $s_j$ denote the original position of the $j$th predictor produced by $Z(s)$ and the observed data is randomly permuted version of the original data, i.e., $x_{ij} = Z_i(s_j)$.

**Case 1** There is a underlying true ordering of the predictors, i.e., there exists a permutation $\{\pi(1), \cdots, \pi(p)\}$ of $\{1, 2, \cdots, p\}$ such that $s_{\pi(1)} < s_{\pi(2)} < \cdots < s_{\pi(p)}$ is the true, but unknown, ordering of the predictors. If such ordering exists, we can take $m = 1$ and the predictors can be projected on $[0, 1]$ via UDS. Let $\{\hat{s}_1, \cdots, \hat{s}_p\}$ be the estimated locations of the $p$ predictors on $[0, 1]$ obtained via UDS. Let $\{\psi(1), \cdots, \psi(p)\}$ be the permutation of $\{1, 2, \cdots, p\}$ that orders $\{\hat{s}_1, \cdots, \hat{s}_p\}$. Then, under some regularity conditions $\psi(j) = \pi(j)$, $1 \le j \le p$, $\forall p$. Thus, UDS can correctly identify the true relative pairwise distances among the predictors. For proof, see ref. [27].

**Case 2** Suppose the ordering does not exist, e.g., suppose multiple predictors are equidistant from one another. Clearly, $m = 1$ may not be a valid assumption and results corresponding to Case 1 become untenable in this situation. For the second-order approximation, we start with $m = 2$, i.e., we would like to obtain the location of the predictors in a compact subset of $\mathbb{R}^2$. Without loss of generality, we project the locations on unit square ($[0, 1]^2$). Let $d_{jk}$ be the observed distance between the $j$th and the $k$th predictor in the higher dimension and $\delta_{jk}$ be their true, but unobserved, distances in the 2D plane. Under the assumption of Euclidean metric, $\delta_{jk} = \sqrt{\sum_l (s_{j,l} - s_{k,l})^2}$, where $s$ is now 2D coordinate system denoting the true location of the predictors $j$ and $k$ in unit square. As in Case 1, we can assume that $\pi(\cdot)$ is the underlying true permutation of 2D configurations of the $p$ predictors. Our goal is to draw inference on the locations of each predictor, i.e., estimate $s_j \in [0, 1]^2$.

Oh et al.[44] developed a Bayesian estimation procedure to estimate $s$, based on observed distance by assuming $d_{jk} \sim N(\delta_{jk}, \sigma^2)I(d_{jk} > 0)$ at the data level. For the location process, we specify a spatial homogeneous Poisson process (HPP) with constant intensity $\lambda = p/[0, 1]^2$ that essentially distributes locations of $p$ predictors randomly in an unit square. Since this corresponds to complete spatial randomness, an alternative specification of location process is given by $\boldsymbol{s} = \{s_1, s_2, \cdots, s_p\} \sim \mathrm{U}([0, 1]^2)$, U denoting the uniform distribution[45]. Note that, the properties of HPP guarantee that the HPP operating on the unit square can be further partitioned into disjoint cells and the entire location process can be expressed as the superposition of the HPPs operating on these disjoint cells. Furthermore, as the volume of each cell (within the unit square) goes to zero, so does the probability of observing more than one event in that cell[46]. We also note that, we do not assume that the location process shows any clustering tendency a priori; however, the uniform specification is flexible enough to capture clustering a posteriori[45]. Let us denote the set of observed and true distances by $\boldsymbol{d}$ and $\boldsymbol{\delta}$, respectively. Our data model is then given by

$$f(\boldsymbol{d}|\boldsymbol{s}, \sigma^2) \propto (\sigma^2)^{-\frac{q}{2}} \exp\left[-\frac{1}{2\sigma^2}\sum_{j>k}\left(d_{jk} - \delta_{jk}\right)^2 - \sum_{j>k}\log\Phi\left(\frac{\delta_{jk}}{\sigma}\right)\right], \quad (10)$$

where $q = \binom{p}{2}$ is the total number of distances in the dataset and $\Phi(\cdot)$ is the usual standard normal CDF. At the process level, we have

$$\boldsymbol{s}|p \sim \mathrm{U}([0, 1]^2). \quad (11)$$

Finally, the prior is given by $\sigma^2 \sim \mathrm{IG}(a, b)$ with $a > 2$, $b > 0$, and IG denoting the inverse gamma distribution. Consequently, the full posterior distribution is given by

$$[\boldsymbol{s}, \sigma^2|\boldsymbol{d}] \propto (\sigma^2)^{-\left(\frac{q}{2}+a+1\right)}\exp\left[-\frac{1}{2\sigma^2}\sum_{j>k}\left(d_{jk} - \delta_{jk}\right)^2 - \sum_{j>k}\log\Phi\left(\frac{\delta_{jk}}{\sigma}\right) - \frac{b}{\sigma^2}\right]. \quad (12)$$

When $q$ is large, $\sum\log\Phi(\cdot) \approx 0$, the full conditional posterior of $\sigma^2|\cdot$ is approximated by $\mathrm{IG}(\frac{q}{2}+a, \frac{1}{2}\sum_{j>k}(d_{jk} - \delta_{jk})^2 + b)$. However, the conditional posterior of $s$ is not available in closed form, a Metropolis-in-Gibbs sampler is used to obtain posterior realizations of the locations. Since $s$ are identifiable only up to an automorphism, convergence of the Markov Chain Monte Carlo is assessed on

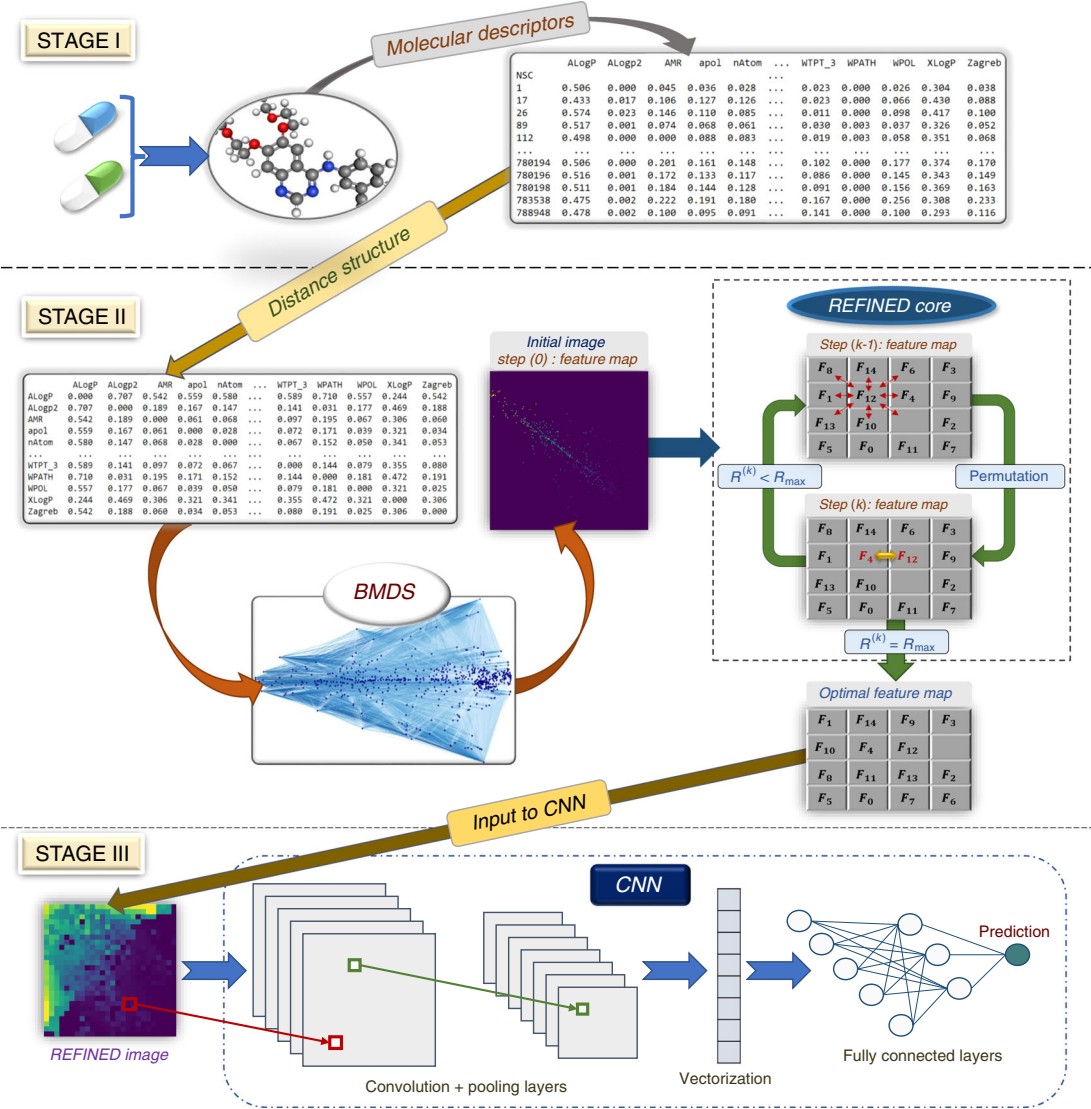

**Fig. 5 Pictorial representation of the REFINED approach.** Overview of REFINED-CNN methodology for a representative application of drug sensitivity prediction using high-dimensional input features, such as molecular descriptors of drugs or genomic profiles of cell lines. STAGE I, calculate the pairwise dissimilarity matrix for the input features (672 × 672 Euclidean distance matrix for PaDEL descriptors of ~52,000 unique drugs in NCI60 here). STAGE II, apply BMDS on this distance matrix to generate an initial image (of size 26 × 26 here) and apply hill climbing to arrive at an optimal configuration, i.e., the REFINED image, by maximizing the similarity between initial and final dissimilarity matrices. STAGE III, train a suitable CNN architecture with the REFINED images and predict sensitivity for a new sample (a given drug here).

$\delta$ and $\sigma^2$. Furthermore, following the recommendation of ref. [44], we used the posterior mode of $s$ as the point estimate of the predictor location.

Once these locations are estimated, we have a set of point-referenced predictor location in the domain of interest. The domain is then subjected to regular square tessellation such that each pixel contains at most one location—essentially rewriting the point process a posteriori as superposition of constant intensity local point processes operating on disjoint cells that the unit square is partitioned into. In addition, the constraint of allowing at most one predictor location per-pixel implies that the posterior location process is approximated by a Non-HPP[46]. The foregoing hill climbing algorithm is then applied to arrive at locally optimal configuration (the location process is not uniquely identifiable. Local adjustment via hill climbing is done to mimic the swapping step of the *partition-around-medoid* algorithm).

Once the tiled surface associated with the feature space is obtained, we have the observed value, $x(s_j)$, for each row of $X$. The intensity at each pixel is, therefore, determined by $x(s_j)$. Pixels that do not contain any predictor location are assigned null values. This pixelated image on unit square is our second-order approximation of the random functions developed in ref. [27]. We can then deploy any suitable smoothing operation (for example, autoregressive spatial smoothing[47]) to generate the corresponding predictor images (such as shown in Fig. 1). Furthermore, each posterior realization of $s$ can be used for data augmentation purpose in CNN architecture. Also, since Euclidean metric is invariant under translation, rotation,

and reflection about the origin, any such perturbation will not affect the relationship between the response and predictors.

Note that, even if there exists ordering among the covariates, we can still generate these images in the following way. Since[27] guarantees that the relative pairwise distances among the predictors, estimated from UDS, are consistent estimators of the true relative distances, we can posit a calibration model for these estimates $\hat{d}_{jk}$ to connect with the true distance, i.e., $\hat{d}_{jk} \sim N(\alpha_0 + \alpha_1 \delta_{jk}, \sigma^2)I(d_{jk} > 0)$.

**Datasets and preprocessing.** To evaluate our framework, we considered three datasets: (a) a synthetically generated dataset, (b) NCI60 dataset consisting of drug responses following application of >52,000 unique compounds on 60 human cancer cell lines[20], and (c) GDSC dataset that contains responses to 222 anticancer drugs across ~972 cancer cell lines with known genomic information from GDSC[21]. In scenario (b), we use the chemical descriptors of drugs to predict drug responses in a specific cell line. In scenario (c), we consider two heterogeneous predictor set—(i) gene expressions for cancer cell lines and (ii) chemical descriptors for applied drugs, and use both these type of predictors to predict drug responses.

We simulated a synthetic dataset with $P$ correlated features for $N$ samples, where for each simulation 20, 50, and 80% of the features were spurious. The features were simulated from a zero-mean Gaussian process with stationary isotropic covariance matrix whose $(i, j)$th element is given by $\gamma^{|i-j|}$. We define

$P = 20, 50, 100, 400, 800, 1000, 2000, 4000$ and for each $P$, the number of simulated samples is, $N = 50, 200, 600, 1000, 2000, 5000, 10000$. We simulated the target values by simply multiplying random weights to the features. For example, $N$ target values with 100 features ($X_{N \times 100}$) with 20% spurious features were generated using the relation $X[\mathbf{w}_r, \mathbf{w}_0]^T$, where $(\mathbf{w}_r)_{80 \times 1}$ are nonzero random weights and $(\mathbf{w}_0)_{20 \times 1}$ are zeros.

The US National Cancer Institute (NCI) screened >52,000 unique chemicals on ~60 human cancer cell lines. The chemical (drug) response is reported as $GI_{50}$ (in molar, M) that is the concentration required to achieve 50% of maximal inhibition of cell proliferation[20]. All the chemicals have an associated unique NSC identifier number that is assigned to identify agents when they are submitted for clinical trials to the Cancer Therapy Evaluation Program. We used the NSC identifiers to obtain the chemical descriptor features and then used PaDEL software[34] to extract these features for each one of the chemicals. The descriptors with >10% zero or missing values were discarded. The final dataset consists of 52,126 chemicals, each with 672 descriptor features and 59 cancer cell lines. To incorporate the logarithmic nature of dose administration protocol, we calculated the negative-log concentration of $GI_{50}$ values, termed as NLOGGI50 (for instance, NLOGGI50 = 8 represents $GI_{50} = 10^{-8}$ M or 10 nM). The drug response distribution for two illustrative cell lines are shown in Supplementary Fig. 6a, b. We selected 17 cell lines with >10,000 drugs, to ensure availability of enough data points for training deep learning models. The number of drugs applied on each selected cell line are provided in Supplementary Table 11. All these preprocessing steps were conducted in the training phase.

The GDSC project is a collaborative effort between the Cancer Genome Project at the Wellcome Sanger Institute (UK) and the Center for Molecular Therapeutics at the Massachusetts General Hospital Cancer Center (USA). According to the GDSC website, this collaboration aims to incorporate the expertise at both sites for discovering biomarkers that can be used to identify the most suitable candidates for potential anticancer therapies. We have two different sets of input for GDSC v7.0— (i) PaDEL descriptors for the 222 available drugs applied on ~972 cancer cell lines, and (ii) microarray gene expression data for those 972 cell lines before drug application. The response consists of experimentally obtained $IC_{50}$ values for a particular drug applied on a particular cell line, therefore, each sample now is a unique drug–cell line combination. Considering 222 drugs applied on almost 972 cell lines, the total number of available samples becomes ~177,000.

Since in prior studies[1,4,48], an initial feature selection was used to reduce the feature size before training the predictive model, we also used an a priori RELIEF-F feature selection[49] in scenario (c) for easier comparison with earlier studies. Since multiple drugs were tested on each cell line, a common subset of 1211 genes in the top 8048($\approx$8000) genes for all 222 drugs were selected for each drug. Similar to (b), we once again used PaDEL to obtain the initial chemical descriptor set, and the descriptors with >10% zero or missing values were discarded resulting in a final set of 992 descriptors for each drug. All gene expression and drug descriptor values were normalized in [0, 1].

**Predictive models**. We used the REFINED images to train a CNN for drug sensitivity prediction purposes in the three abovementioned scenarios. We compared the performance of the REFINED-CNN with various standard ML models, such as EN, RF, SVR, and deep ANN in the regression cases. For the classification case involving NCI60, we also trained a LR model along with the abovementioned model classifiers. In addition, for the scenarios (b) and (c), we also trained two other CNNs with images created by random 2D projection matrix (Random-CNN), and images created using PCA coordinates (PCA-CNN). The details of these approaches are described in "Alternative image generation approaches" in the "Results" section.

Note that for the synthetic dataset and NCI60, the predictors are a vector of real values (chemical descriptor values for NCI60) being converted to images. For the GDSC dataset, we have two types of input features—chemical descriptors describing the drugs and gene expression describing the cell lines. We generated individual images for each feature type and used both of them as input to the CNNs. For the RF, SVR, ANN, and EN, these two types of features were appended and used as the predictors. We trained REFINED-CNN and the other competing models on same set of samples, where each sample is a combination of one drug tested on one cell line. All the models were tested on a separate set of the same samples.

The distribution of NLOGGI50 shows a massive point mass (Supplementary Fig. 6), indicating that an overwhelming majority of drugs are not sensitive for majority of the NCI60 cell lines. Thus, we considered a classification problem to identify whether a drug is sensitive or resistant for the NCI60 cell lines. For selecting the sensitivity threshold, we used the observed normalized $\log GI_{50}$ (NLOGGI50) distribution for different cell lines and empirically located the threshold at 4.25. All the drugs with NLOGGI50 < 4.25 were considered resistant and the rest are considered as sensitive.

**Convolutional neural network**. CNNs are designed to model multidimensional arrays, where convolutional layers along with pooling layers are adaptive feature extractors connected to sequential fully connected (dense) layers[6]. A convolutional layer consists of multiple kernels connected to a local path of neurons in the previous layer, where all neurons share same parameters to generate a feature map. Thus, all neurons within the feature map scan same features in different locations of the previous layer. The pooling layer summarizes the feature map by finding the maximum or mean of each adjacent kernel that reduces the number of model parameters[5]. We used two different CNN architectures—a sequential CNN for modeling the NCI60 and synthetic dataset, and a hybrid CNN that can accommodate drug and genetic characterization images for the GDSC datasets.

The sequential CNN regressor contains six learned layers: one input layer, two convolutional layer, two dense layer, and one output layer. The CNN input dimension is same as the input image dimension, $26 \times 26$. The convolutional layer contains 64 $7 \times 7$ kernels convolving with valid border mode and stride = 2, followed by batch normalization and rectified linear unit (ReLU) activation layer[50]. Each dense layer is followed by a batch normalization layer and ReLU activation layer. Number of neurons of the dense layers are 256 and 64, respectively. A dropout layer with retaining probability of 0.7 was added before the output layer. The above architecture remains same for Random-CNN, PCA-CNN, and REFINED-CNN.

The sequential CNN classifier contains input layer with the same size as the CNN regressor, three convolutional layer with 16 $7 \times 7$ kernels, 32 $7 \times 7$ kernels, and 64 $3 \times 3$ kernels, respectively. Each convolutional layer is followed by a batch normalization[51] and a ReLU layer. The third ReLU layer is followed by two dense layers with 256 and 64 neurons, respectively. Same as the CNN regressor, each dense layer is followed by a batch normalization, ReLU and a dropout layer. The CNN classifier architecture remains same for Random-CNN, PCA-CNN, and REFINED-CNN. We used Adam optimizer to train both CNN regressor and classifier. The CNN regressor and classifier architecture is shown in Supplementary Fig. 21.

We used hybrid CNNs with two input sets to model GDSC dataset, where two separate sets of images are used as input to the two arms of the CNN. Each arm contains three convolutional layers where the last convolutional layers are concatenated, and followed by two sequential dense layers. The two input layers represent the cell lines and drug images that defines each convolutional layer dimension. The three convolutional layer of each arm have 60 $5 \times 5$ kernels with stride = 1, 72 $6 \times 6$ kernels with stride = 2, and 72 $5 \times 5$ kernels, respectively. The last convolutional layer of the arm that takes drug images as input has stride = 1 and the other arm that takes cell lines images as input has stride = 2. Each convolutional layer is followed by batch normalization and ReLU activation layers. The last two convolutional arms of the CNN are concatenated and connected to two sequential dense layers with 305 and 175 neurons, respectively. A batch normalization and a ReLU activation layer is included after each dense layer. A dropout layer with retaining probability of 0.7 is placed before the output layer. The CNN model architecture is shown in Supplementary Fig. 16. The hybrid CNN was again trained by an Adam optimizer, and the same architecture was utilized for Random-CNN, PCA-CNN, and REFINED-CNN. There are some variations in the architecture depending on the training set size that are explained in "GDSC sensitivity prediction". All tuning parameters were chosen via a comprehensive search described next.

**Hyperparameter selection**. To tune the competitive models, we did a grid search on the following hyperparameters for each model using hold out for partitioning the data. We randomly partition both NCI60 and GDSC data to 80% training, 10% validation, and 10% test. The same training, validation, and test sets were used for all models. The hyperparameters were tuned using the training and validation sets. The test sets were held out for evaluating the final performance of each tuned model.

- RF: the number of decision trees, $N_{trees} \in (100, 700)$. For the optimal feature size for the best split at each node, we tried all the options provided by scikit-learn[52], including $p$, $\sqrt{p}$, and $\log_2 p$ where $p$ denotes feature size.
- SVM: radial basis function kernel parameter, $\gamma$ set by *scale* and *auto* options of scikit-learn and the regularization parameter, $C \in (0.01, 100)$.
- EN/LR: the regularization term, $\alpha \in (0.3, 0.7)$ and $L_1$ ratio $\in (10^{-6}, 10^{-4})$.
- ANN: number of hidden layers $\in (3, 6)$, number of neurons per hidden layer, and learning rate of the Adam optimizer.

  - Hidden layer 1: #neurons $\in (800, 1200)$.
  - Hidden layer 2: #neurons $\in (600, 1000)$.
  - Hidden layer 3: #neurons $\in (400, 700)$.
  - Hidden layer 4: #neurons $\in (200, 500)$.
  - Hidden layer 5: #neurons $\in (50, 250)$.
  - Hidden layer 6: #neurons $\in (20, 80)$.
  - Learning rate: $\rho \in (10^{-6}, 10^{-3})$.

- CNN: as we did not have access to GPUs, we did not do comprehensive hyperparameter grid search for the CNNs. The current parameters were chosen over about hundreds of run. The hyperparameter space that we searched was number of convolutional layers, number of dense layers, and learning rate of the Adam optimizer. In each convolutional layer, we seek

optimum number of kernels, kernel size, and stride. For each dense layer, we checked multiple numbers of neurons. The insights that we gained from thousands of runs are as follows:

- $7 \times 7$ kernels were better than smaller kernels of size $3 \times 3$ for the convolutional operator. For larger number of features, larger kernels might be desirable.
- Using stride > 1 is more effective in embedded dimensionality reduction than the pooling layer.
- Using two or more sequential convolution layers with kernel size = $7 \times 7$ and stride = 2 reduces the feature map dimension considerably compared to the input image. Therefore, large number of kernels is recommended, at least for the last convolution layer, to provide sufficient number of extracted features for the dense layer.
- The width of network is as important as the depth of the network.
- The Adam optimizer is recommended.

For unbiased evaluation of models, nested cross-validation (NCV)[53] is often considered where an inner CV is used for model selection (hyperparameter selection) and an outer CV is used for evaluating the model tuned by the inner CV. However, NCV is often extremely computationally intensive, and thus we considered a training-validation-test (hold out), where the hyperparameters are tested on the validation set and the selected model is evaluated based on the separate test set. We used this hold-out approach as the sample size is relatively large and thus, both hold out and NCV are expected to provide similar results for comparing different modeling approaches. To illustrate the similar behavior, we compared the results of NCV and hold out using three randomly selected NCI60 cell lines. As the results provided in Supplementary Fig. 22 indicates, the difference in the results are minimal. On the other hand, to compare the time complexity of the two approaches, we selected one cell line with a fixed CNN architecture and defined a grid for hyperparameters search. Within a single run spanning ~48 h on Texas Tech University high performance computing cluster, 50 different models can be tried using hold out, while only four models can be tried using NCV. Given the time complexity and similar performance of the two approaches, we decided to use the three-set hold-out approach for hyperparameter selection and model evaluation, as we could search considerably larger space of hyperparameters while gaining reliable estimate for model performance.

**Reporting summary**. Further information on research design is available in the Nature Research Reporting Summary linked to this article.

## Data availability

NCI60: the $GI_{50}$ data and associated drug chemical information that support the results of this study are available in the Development Therapeutic Program (DTP) repository (https://dtp.cancer.gov/databases_tools/bulk_data.htm) at NCI and the PubChem database (https://pubchem.ncbi.nlm.nih.gov/) at National Library of Medicine (NLM). GDSC: the $IC_{50}$ data and cell line screening data that support the results of this study are available at the GDSC repository (https://www.cancerrxgene.org/downloads/bulk_download) and drug chemical information are available in the PubChem database (https://pubchem.ncbi.nlm.nih.gov/) at NLM. PaDEL: the PaDEL software that was used to convert the drug chemical information to molecular descriptors is available at http://www.yapcwsoft.com/dd/padeldescriptor/.

## Code availability

The source code and scripts used in the paper have been deposited in GitHub (https://github.com/omidbazgirTTU/REFINED).

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

## Acknowledgements

We thank Carlos De Niz, PhD for providing us processed chemical drug descriptors of the NCI60 dataset. Research reported in this publication was supported by the National Institute Of General Medical Sciences of the National Institutes of Health under Award Number R01GM122084. The content is solely the responsibility of the authors and does not necessarily represent the official views of the National Institutes of Health.

## Author contributions

R.P., S.G., and O.B. conceived of the presented idea. O.B., S.R.D., R.P., and S.G., designed the study. O.B., R.Z., and R.R. performed the simulation experiments. R.P., S.G., and O.B. interpreted the analysis results. O.B., S.R.D., R.P., S.G., and R.Z. wrote the manuscript. All authors have read and approved the final manuscript.

## Competing interests

The authors declare no competing interests.
