## [Peer Review File · Nature Communications]

Reviewers' comments:

Reviewer #1 (Remarks to the Author):

In this manuscript, the authors propose the use of a 2D representation of high-dimensional feature vectors as input to CNNs to solve classification and regression problems. They demonstrate the performance of their approach on synthetic data and on the problem of predicting cell line drug response, compared to Random Forests, Support Vectors and Neural Networks.

Although this is an interesting approach and it appears to outperform RF, SV and ANN methods, a deeper look at the manuscript reveals a number of major issues that render the conclusions of this study questionable.

1. REFINE Model

The presentation of the model is very hard to follow. Some points that should be addressed are:

- The authors change notation multiple times. For example, in section 2.1.1: x is defined as $Z(s)$ but 2 paragraphs later (Case 2) x is used again to denote the "true" position, then s is used again ("ie estimate s_j ") and in the very next sentence "estimate $x(\cdot)$ ". In addition, p is used to denote both the number of feature and the rate parameter of HPP
- With respect to the prior of s . It's mentioned initially that it's a Homogeneous Poisson but then a Uniform is specified in eq (2). It's unclear which of the 2 the authors used. In case it's the former, why it doesn't appear on the posterior. The authors claim that HPP essentially forces a constraint of 1 voxel per picture but this is not obvious. The area $A(s)$ is not defined clearly.
- The prior of σ is specified as Inverse Gamma in page 6 and Inverse Gaussian in page 7 (probably by mistake). Then its posterior is approximated using the conjugate form without clear justification.
- There is a typo ($(q=p^2)$). It should be $q=(p^2)$. Also, in the posterior the exponent of σ squared should probably be $-(q/2+a+1)$
- The hill climbing algorithm is not well described either. It's mentioned that its objective function is related to the "true" distance δ but s is already used to define δ so it's not clear how changing them can improve the objective.
- Finally, the sentence "Assuming that the intensity...developed in [26]." is particularly hard to follow for this reviewer.

2. CNN hyper-parameter tuning

The authors mention that tuning parameters for their CNN have been determined via cross-validation. However, there is a lot of missing information and several questions arise:

- Was the cross validation run on the training set alone?
- Was preprocessing also included in the cross validation? More specifically, this refers to data imputation, gene selection (page 9) as well as the mapping of features using REFINE. To prevent data leak, all preprocessing steps should be part of the training phase.
- What was the grid of parameters tested during cross validation?
- How was the actual network architecture decided in the first place? Was it fixed from the very beginning, or were layers added during the study? This can (and maybe should) also be part of the optimization.
- The input size ($64 \times 64 = 4096$) is not particularly large for biological datasets that tend to be wide (eg 20,000 genes like GDSC). Did the authors experiment with bigger sizes?
- Finally, I propose the use of full nested cross validation, instead of the split to training-validation-testing.

3. RF, SV and ANN tuning

In line with comment #2, the same questions apply to the training and tuning of these three learning methods.

4. Other comparisons

The comparison with RF, SV and ANN is a good idea, but elastic nets have also been tried on the

GDSC dataset with relative success. The authors need to compare to this method as well. In addition, the ultimate goal of learning drug sensitivity models from cell line transcriptomic data is to apply these models to patient data and predict their response to drug treatments. One such gold standard dataset is the bortezomib clinical trial on multiple myeloma. Elastic net models, trained on GDSC, have been able to make statistically significant predictions of responders versus non-responders to this drug:

<https://genomebiology.biomedcentral.com/articles/10.1186/gb-2014-15-3-r47>

The authors should test whether their bortezomib model is also equally successful.

Residuals plots, especially those on figure 5, seem very weird. The observed data do not span the whole [0, 1] interval (in figure 6 this is more pronounced). All competing models have very pathological behavior, they seem to under estimate low values and overestimate high ones. ANN also appears to be constant for low values. It looks like just a simple linear term would suffice to fix this behavior and it's surprising that powerful models like these failed to do so.

5. Metrics and statistical significance

The authors should report confidence intervals for all their evaluation metrics. Also, the statistical significance of the difference in performance across methods should be assessed. Finally, for the classification problem, the authors should report AUCs and/or similar metrics.

Moreover, their NRMSE metric is deviance based (it looks like the square root of $1 - R^2$) and thus it's not well suited to compare models (same or different) trained on different datasets (see table 3).

For figure 4, the authors don't specify which quantity is represented by the color-scale.

Finally, bias is not a standard metric because residual plots typically do not have clear constant slopes.

6. 2D representation of feature vectors

The main idea of this study is that the organization of unordered feature vectors into 2D image representations can confer an advantage in learning tasks. However, this claim has not been tested in detail. The authors should test the following and possibly come up with more ideas of their own:

- Input randomly-ordered 2D image representations into their CNN architecture and retrain (using nested CV)
- Study the performance as a function of the number of input features and again compare to randomly ordered 2D images
- Compare to other approaches that create 2D representations from unordered vectors

7. Other comments

- It's not clear how the random dataset was created. Which distributions were used? Were the target variables normalized as well?
- Page 8: Normalized IC50s to the [0,1] interval. It is not clear if this was done globally or per drug, or per cell line. Also, figures 5, 6, and 7 to span the whole range. Please clarify. Why is this necessary? I am not sure this is justified.
- Page 8: Figure 3c. The title says "IC50" but the legend says "NLOGIC50".
- Page 16: "we used the NCI60 coordinates to generate the drug images for the 222 drugs"; it appears that this provides an unfair advantage to the REFINED algorithm over the rest of the methods.
- Page 10: The hybrid CNN receives inputs that are pre-organized into the two distinct data types (transcriptomics and chemical descriptors). Again, this seems to be unfair to the other algorithms. To avoid this, please also compare the performance of all algorithms based on transcriptomic data alone
- I apologized if I missed this in the methods, but it is not clear if different models are trained per drug, or whether there is a single model (e.g. a multi-class multi-label classification). To avoid unfair comparisons, all tested algorithms should be trained to the same learning task.
- Finally, I suggest that the authors come up with more creative ways to present their results in the main figures. Most of the presented results figures and tables are good and informative but are

better suited for the Supplemental section (with the exception of figures 2 and 4). The main figures should visualize the information better. Also, figure 1 is very important, but needs to be improved.

Reviewer #2 (Remarks to the Author):

This paper proposes the method called REFINED, which represents multidimensional feature vectors as 2D image forms to exploit Convolutional neural networks (CNNs) as CNNs achieved human-level performance in image-based tasks. It applies Bayesian Multidimensional Scaling (BMDS) and hill-climbing algorithm for mapping each feature into an optimal location on a 2D image space. The paper demonstrates its advantages: automated feature selection without explicit feature selection techniques, an increase in predictive accuracy, and bias correction.

The approach of REFINED method is interesting as it converts feature vectors to 2-dimensional locations reflecting feature similarity. Despite that, this paper has critical points that compromise contributions the paper asserts.

1. The presentation is fairly poor.

Above all, 'automated feature selection' is a misleading term. The authors emphasize automated feature selection as a core contribution of the paper, however, 'feature selection' seems incorrect. Instead, 'automated feature extraction' is an appropriate term since it is derived from CNNs that allow feature extraction.

Moreover, the authors did not elaborate specific details of how each feature value is visualized in the generated 2D image.

In addition, some fundamental notations have an incorrect definition in section 2.1.1. For example, in equation (1), q must be defined as a combination of p taken 2 because q denotes all possible pairs. Likewise, IG denotes Inverse Gaussian distribution in this paper, but it should refer to Inverse Gamma distribution which is theoretically correct. Besides, some notations are not specified. These could be considered as an incomplete understanding of the theoretical backgrounds.

2. In drug response tasks, three baseline models are conventional which do not offer state-of-the-art performance. As the authors imply REFINED-CNN provides better performance than others, the authors should assess the performance of REFINED-CNN comparing with other advanced models giving a state-of-the-art performance (i.e., Deep-Resp-forest, Transfer learning-based, Heterogeneous graph)

3. This paper does not clarify how components in the method contribute to the advantages REFINED gives. REFINED is composed of Bayesian MDS and hill-climbing algorithm which are global dimensional reduction and local search optimization, respectively. However, the authors do not explain specific reasons for using BMDS instead of other global dimension mapping methods such as Isomap. Also, it lacks a comparison of local search and local dimension reduction methods (e.g., Laplacian Eigenmaps, LLE). Hence, Ablation tests are desired.

** Minor comment **

1. In section 2.1.1, a constraint is omitted in the notation. For instance, there should be a constraint of $j \neq k$, $j, k = 1, 2, \dots, p$ in estimating $x(\cdot)$ based on observed distance. Also, notation $S_{j,l}$ (in 4th paragraph, page 6) is undefined.

2. Analyses are inadequate. There is no qualitative analysis that examines the effectiveness of the REFINED-CNN method. On top of that, quantitative analyses only describe results, neither

interpretation nor intuition. In-depth analyses, such as case study and error analysis would increase the overall quality of the paper.

3. Using accuracy as the only metric in classification tasks is inappropriate. It could mislead an evaluation when an output distribution is imbalanced. F1 score would be better than the single accuracy.

4. Are there any reasons to utilize a hill-climbing algorithm despite its local optima problem?

5. Thorough details of baseline models should be provided.

6. Figure1 does not reflect its caption. The figure illustrates GDSC dataset, whereas its caption is about NCI60 dataset.

7. Figure 1 has a low resolution.

8. 'Voxel representation' appears to be inaccurately used with 2D image space, since voxel indicates 3-dimensional space.

Detailed Response to Reviewer Comments (“REFINED (REpresentation of Features as Images with NEighborhood Dependencies): A novel feature representation for Convolutional Neural Networks”)

Major points: *The authors would like to thank Reviewer No. 1 for his/her constructive review of the manuscript and their valuable comments. With regard to the concerns raised by this Reviewer, the following changes have been incorporated in the revised version:*

“1. REFINE Model

The presentation of the model is very hard to follow. Some points that should be addressed are:

- The authors change notation multiple times. For example, in section 2.1.1: x is defined as $Z(s)$ but 2 paragraphs later (Case 2) x is used again to denote the “true” position, then s is used again (“ie estimate s_j ”) and in the very next sentence “estimate $x(.)$ ”. In addition, p is used to denote both the number of feature and the rate parameter of HPP”

We have completely revised the section and made the notations coherent. Our intention in the above specification was to bring out the fact that we are randomly distributing “ p ” points (locations associated with “ p ” features) in unit square with a constant intensity function “ $p/\text{Area of unit square}$ ”. We have revised the text to make the issue clear.

“- With respect to the prior of s . It’s mentioned initially that it’s a Homogeneous Poisson but then a Uniform is specified in eq (2). It’s unclear which of the 2 the authors used. In case it’s the former, why it doesn’t appear on the posterior. The authors claim that HPP essentially forces a constraint of 1 voxel per picture, but this is not obvious. The area $A(s)$ is not defined clearly.”

As mentioned above, HPP is equivalent to complete spatial randomness. Since we are only distributing “ p ” locations, an equivalent specification is to generate “ p ” 2-dimensional coordinates from Uniform distribution operating on unit square (Chandler & Royle, 2013).

*-“ The prior of σ is specified as Inverse Gamma in page 6 and Inverse Gaussian in page 7 (probably by mistake). Then its posterior is approximated using the conjugate form without clear justification.
- There is a typo ($q=p-2$). It should be $q=(p-2)$. Also, in the posterior the exponent of σ squared should probably be $-(q/2+a+1)$ ”*

We thank the reviewer for pointing that out. We have rectified the statements.

- “The hill climbing algorithm is not well described either. It’s mentioned that its objective function is related to the “true” distance δ , but s is already used to define δ so it’s not clear how changing them can improve the objective.”

It is to be noted that the posterior realizations of s_i are not identifiable and the convergence of MCMC cannot be monitored for s_i ’s. Rather the convergence is monitored for the δ and σ chains (On & Raftery, 2001). The Hill Climbing is done as a post-processing step to see if local adjustments of locations can improve the stress function. This step is done to mimic the swap step associated with Partition-around-medoid algorithm. More formally, observe that the output of the BMDS algorithm produces a set of locations \hat{s} in the unit square. This is not a veritable image. To convert this collection of points and their associated values (value of the predictor) into an image, we need to impose a grid on unit square such that at most one \hat{s}_i is contained in each grid. The hill-climbing operation is

undertaken to identify the coarsest resolution of this grid. More specifically, suppose \hat{s}_i is contained in grid i and \hat{s}_j is contained in grid j . Let $\widehat{\delta}_{ij}$ be the BMDS posterior estimate of the distance between \hat{s}_i and \hat{s}_j , the hill-climbing operation is performed to minimize the distortion associated with placing \hat{s}_i and \hat{s}_j in their respective grid –centers, i.e, $\text{dist}(\text{center of grid } i, \text{center of grid } j) \approx \widehat{\delta}_{ij}$. Clearly, given the global optimizer producing $\widehat{\delta}_{ij}$, the hill-climbing operation identifies a set of local adjustments that closely follows the global optimum estimates.

- “Finally, the sentence “Assuming that the intensity...developed in [26].” is particularly hard to follow for this reviewer”.

We have revised that statement to make it clearer.

“2. CNN hyper-parameter tuning

The authors mention that tuning parameters for their CNN have been determined via cross-validation. However, there is a lot of missing information and several questions arise:

- Was the cross validation run on the training set alone?*
- Was preprocessing also included in the cross validation? More specifically, this refers to data imputation, gene selection (page 9) as well as the mapping of features using REFINE. To prevent data leak, all preprocessing steps should be part of the training phase.”*

As suggested by the reviewer, the preprocessing was conducted on the training phase but was not clarified in the manuscript. We have now revised the manuscript to clearly mention that preprocessing was done in the training phase.

“- What was the grid of parameters tested during cross validation?

- How was the actual network architecture decided in the first place? Was it fixed from the very beginning, or were layers added during the study? This can (and maybe should) also be part of the optimization.”*

As pointed out by the reviewer we have clarified the hyper-parameter search task for all the models under the “2.3.2 Hyper parameter search” section of the revised manuscript. We have included the range of values for the hyper-parameters of each model that were used in the hyper-parameter search. We also explained some of the insights that we obtained while conducting the hyper parameter search for the REFINED CNN.

We did not conduct a grid search on the test data. The test set data was hold out for only reporting performance of each model, and we used the same test set for all the models.

“- The input size (64x64 = 4096) is not particularly large for biological datasets that tend to be wide (eg 20,000 genes like GDSC). Did the authors experiment with bigger sizes?”

As recommended by the reviewer, we conducted further experiments with larger number of genes on the GDSC dataset that leads to creating larger REFINED images. Overall, we used multiple set of gene sizes including: 1211, 2147, 2985, 4271, and 8048 genes, which are explained in the GDSC results section of the manuscript as below:

“We also incorporated larger number of genes to experiment the impact of creating larger images using REFINED. We selected larger common subset of genes including (2147 ~

2000, 2985 ~ 3000, 4271 ~ 4000, 8048 ~ 8000) genes and created the REFINED images to train the same CNN architecture that we used to train the GDSC data with 1211 selected genes. We also trained other competing models with the same hyper parameters. In our experiment, the SVR model crashed after several hours due to the larger memory requirement while estimating the kernel by computing the distance function between each point in the dataset. Thus, we used a bag of SVRs, where we selected subset of 400 features (appended gene expression and drug descriptors) to train 100 SVRs in parallel. The complete results are provided in figure 33 and table 18 along with the corresponding confidence intervals in figure 34 35 36 of the appendix. We observed that the trend of REFINED CNN outperforming other approaches is maintained irrespective of the number of genes used for training the models.”

We have also modified the synthetic data section, where now the number of features that we generate range from 20 to 4000, and the number of samples range from 50 to 10000 for each set of generated features. The following paragraph has been included in the synthetic dataset section:

“We simulated a synthetic dataset with P correlated features for N samples, where for each simulation 20%, 50% and 80% of the features were spurious. The features were simulated from a zero mean Gaussian process with stationary isotropic covariance matrix whose $(i; j)$ th element is given by $\gamma^{|i-j|}$. P ranged from 20,50,100,400,800,1000,2000,4000 and for each P , the number of simulated sampled ranged from 50,200,600,1000,2000,5000,10000.”

“- Finally, I propose the use of full nested cross validation, instead of the split to training-validation-testing.

3. RF, SV and ANN tuning

In line with comment #2, the same questions apply to the training and tuning of these three learning methods.”

We appreciate the Reviewer’s recommendation on using nested cross validation as it is often the optimum method for hyper parameter search and unbiased evaluation of a machine learning model. However, nested cross-validation can be extremely time consuming when the dataset is large and numerous types of comparisons for different approaches have to be compared as is applicable to our scenario. Thus, we compared the hyper-parameter search performance of Nested Cross Validation against training-validation-testing (hold-out) approach using a random subset of three different cell lines from the NCI60 dataset. For the hold-out approach, we trained the models on the training set, fine-tuned the hyper parameters using the validation set, and evaluated them on the test set. We observed that the trends in terms of performances of different models (as shown in Figure 18 of revised manuscript) remains similar for both nested CV and hold-out approaches and thus the results for the other cell lines and dataset are reported using the less computationally intensive training-validation-testing (hold-out) approach. Our conclusion of similar results for nested CV and hold-out for large datasets has been observed earlier as concluded in <http://www.jmlr.org/papers/volume11/cawley10a/cawley10a.pdf> that the effect of bias in model evaluation using nested CV compared to other approaches decrease as the datasets gets larger.

“4. Other comparisons

The comparison with RF, SV and ANN is a good idea, but elastic nets have also been tried on the GDSC dataset with relative success. The authors need to compare to this method as well. “

As suggested by the reviewer, we included elastic net results for not only GDSC dataset, but also for the NCI60 dataset. Our proposed REFINED-CNN approach outperformed Elastic Net for all considered cases.

“In addition, the ultimate goal of learning drug sensitivity models from cell line transcriptomic data is to apply these models to patient data and predict their response to drug treatments. One such gold standard dataset is the bortezomib clinical trial on multiple myeloma. Elastic net models, trained on GDSC, have been able to make statistically significant predictions of responders versus non-responders to this drug:

<https://genomebiology.biomedcentral.com/articles/10.1186/gb-2014-15-3-r47>

The authors should test whether their bortezomib model is also equally successful.”

As recommended by the reviewer, we did test our approach on the Bortezomib model. However, detailed reading of the supplementary information and codes of the paper elucidated some issues such as use of test data to select optimal cutoff and thus, we didn't include the comparison results in the revised paper. We have created a separate 7-page report (attached at the end of this response to Referees Letter) that provides the detailed results of the bortezomib model comparison.

“Residuals plots, especially those on figure 5, seem very weird. The observed data do not span the whole [0, 1] interval (in figure 6 this is more pronounced). All competing models have very pathological behavior, they seem to underestimate low values and overestimate high ones. ANN also appears to be constant for low values. It looks like just a simple linear term would suffice to fix this behavior and it's surprising that powerful models like these failed to do so.”

As recommended by the reviewer, we corrected the residual plots to represent the non-normalized range. The residuals are plotted in the same manner as previous studies such as :
<https://bmcbioinformatics.biomedcentral.com/articles/10.1186/s12859-018-2060-2>

As shown in figure 3, all the cell lines of the NCI drug response have a massive peak which represents non-responsive drugs and there is limited response lower than that peak and thus, the observed data do not span the whole range [0,1] (initial submission), and [0,8.5] (revised submission). The behavior of ANN is potentially caused by the prevalence of point-mass. In fact, any mean field model will struggle to accommodate point masses. CNN (or other local smoothers, for example Lowess) are essentially defined on highly resolved bins and hence are adept at handling such point masses. We have observed that RF can be modified to such scenarios too when its classifier properties are exploited (<https://www.nature.com/articles/s41598-017-11665-4>). Note that models such as RF are known to have biases that overestimates low values and underestimates high values (PMID: 29589559).

“5. Metrics and statistical significance

The authors should report confidence intervals for all their evaluation metrics. Also, the statistical significance of the difference in performance across methods should be assessed. Finally, for the classification problem, the authors should report AUCs and/or similar metrics.

Moreover, their NRMSE metric is deviance based (it looks like the square root of $1 - R^2$) and thus it's not well suited to compare models (same or different) trained on different datasets (see table 3).”

As suggested by the reviewer, we have reported other metrics for both classifier and regression models.

We report precision, recall, F1-score and AUROC for the classification and NRMSE, Correlation Coefficient and bias for regression. For data augmentation comparison in Table 3, we have added additional metric of normalized mean absolute error for comparison. For the model comparisons, we have kept the same training and testing datasets. We also report statistical significance for both regression and classification tasks. The classification results along with McNemar’s test for statistical significance for NCI dataset are reported in tables 6-8 of the appendices. To report the statistical significance of the difference in performance of the regression task across considered methods (REFINED CNN, Random CNN, PCA CNN, RF, SVR, ANN, EN), we considered the design of a null distribution and compared our results in comparison to the null distribution similar to the approach considered in <https://www.nature.com/articles/nbt.2877>. Random predictions were generated from the training CDF of Y. Under the assumption of the simplest intercept-only model the conditional distribution $Y|X$ is proportional to the marginal distribution of Y. Evaluation of the comparison metrics for each realization of random predictions produces a null distribution of the comparison metric. Similar distribution of the foregoing metrics can be obtained by fitting the candidate models of bootstrapped samples. Therefore, we have two distributions of each comparison metric. Significant lack of predictive power is indicated if distribution of a metric obtained under a candidate model overlaps substantially with the distribution of the corresponding metric obtained under null model. However, there is no observed “test statistic” to perform a formal test. Consequently, we “bag” the comparison metric scores (say NRMSE) coming from the candidate model and the null model and ask the question if there exist sufficient statistical evidence to of the existence of two clusters in a completely unsupervised situation. A formal answer to assess significance is such a situation is given by the Gap Statistic (Tibshirani et al, 2001). This statistic is most powerful in determining $H_0: m=1$ cluster vs $H_1: m=$ more than 1 cluster. Non-rejection of H_0 indicates lack of predictive power for the corresponding candidate model. Furthermore, Gap performs a greedy search to identify suitable number of clusters. In an ideal situation the Gap statistic will identify $m=2$ clusters and the cluster membership will show that members of one cluster contains all the scores coming from the null model while the other cluster contains all the scores from the candidate model. This will not indicate sharp delineation of the candidate model from the null and high level of predictive precision. Cases with $m>2$ is ambiguous and can happen under severe heteroskedasticity. Subsequently, the null model results were paired with each model and the Gap statistics were utilized to report the significance of the difference in performance across the models. We observed that our proposed REFINED CNN performs better than the other 6 competing methods for both average error and statistical performance. The Gap statistics results are provided as tables 12 and 17 and figures 19, 30, 31 and 32 of the Appendix. The following paragraph is included in the revised manuscript to provide the details of the utilized Gap Statistics approach:

“We used Gap statistics [41] to report the significance of the difference in performance across methods of the regression tasks. We paired each model with a null model [1] with bootstrap sampling, where the bootstrap sampling was done on the dose-response values of the test set along with their corresponding predicted values for each model and the null model randomly predicted dose-response values for the sampled test set using the distribution of the training set dose-responses. The process is repeated for 10,000 times and a distribution of NRMSE, PCC and Bias is made for each model along with the null model. The distributions per each metric for each model is paired with null models’ metric distribution. Then cluster centroids are calculated using k-means ($k = 2$) clustering, after the Gap statistics shows appropriateness of using 2 clusters. The difference between each model and the null model cluster centroids per metric represents the difference between them.”

To report the confidence interval of the regression models, we have provided bar plots in figures 20, 21, 22, 34, 35, 36 of the appendix, using a pseudo Jackknife-After-Bootstrap approach as below:

“We calculated 95 % confidence interval for each metric that is used to measure the performance of the methodologies in modeling NCI60 and GDSC dataset using a pseudo Jackknife-After-Bootstrap confidence interval generation approach [4, 31]. Multiple Bootstrap sets were selected from the test samples and error metrics generated resulting in a distribution for each error metric which was used to calculate the confidence interval for a given cell line for NCI60 dataset or a given pair of cell line and drug for the GDSC dataset.”

We have also conducted a robustness analysis (similar to the approach followed in <https://www.nature.com/articles/nbt.2877>) on the regression results for both NCI60 and GDSC datasets as shown in tables 11 and 16 of the revised manuscript. We observed that REFINED CNN outperformed other models more than 90% of the times using the NCI60 dataset. REFINED CNN also outperformed shallow models as well as DRF and HGNN in more than 90% of the times, and the deep models in more than 60% of the times using the GDSC dataset.

The description of the robustness analysis is included in the revised manuscript as:

“In addition to the Gap statistics, all models were subjected to a robustness analysis [1], where we calculate how many times the REFINED CNN outperforms other competing models in 10,000 repetition of bootstrap sampling process.”

The 95% confidence interval for each of the metric for measuring the classifier performance was calculated using the Binomial proportion confidence interval [<https://onlinelibrary.wiley.com/doi/abs/10.1002/sim.4780120902>]. To report the statistical significance of the difference in performance across classification tasks, we used the McNemar test (results in table 8 of the Appendix) with the following details included in the results section of the revised manuscript:

“To compare the statistical significance of the difference in performance between competing classifiers and REFINED CNN, we used the McNemar’s test which is a paired nonparametric statistical hypothesis test. McNemar’s test evaluate whether two models disagree in the same way or not. To compare classifiers with REFINED CNN classifier pairwise, a contingency table is formed and McNemar’s test is applied [40]. The null hypothesis is whether two classifiers disagree by the same amount. Therefore, if the p-value is smaller than a threshold (0.05), null hypothesis is rejected, and the conclusion is: there is a significant disagreement between the two classifiers. The results of comparing REFINED CNN with other models using McNemar’s test is provided in table 8 of the appendix. We observe that the REFINED CNN classifier performance is significantly different than the other classifiers (LR, RF, SVM, ANN, Random CNN, PCA CNN). “

“For figure 4, the authors don’t specify which quantity is represented by the color-scale.”

We thank the reviewer for pointing that out. We have included the quantity “Difference in NRMSE” for the color-bar.

“Finally, bias is not a standard metric because residual plots typically do not have clear constant slopes.”

As pointed out by the reviewer, Bias is not a standard metric, but often residual plots show a clear slope for some well-known models such as RF which is known to underestimate higher values and

overestimate lower values as shown in Figures 9 and 10 of the revised manuscript. We kept residual slope as an additional measure to see whether REFINED CNN is able to solve this problem of underestimating higher values and overestimating lower values. If a model does not possess any bias towards higher or lower value predictions, the slope of the residual plot is expected to be closer to zero.

We observed that REFINED CNN does perform better in terms of residual slope as compared to other approaches. We also applied bias correction to all models and observed that REFINED CNN outperforms others in terms of bias both with or without bias correction as shown in Figure 9 of revised manuscript.

“6. 2D representation of feature vectors

The main idea of this study is that the organization of unordered feature vectors into 2D image representations can confer an advantage in learning tasks. However, this claim has not been tested in detail. The authors should test the following and possibly come up with more ideas of their own:

- Input randomly ordered 2D image representations into their CNN architecture and retrain (using nested CV)*
- Compare to other approaches that create 2D representations from unordered vectors”*

As recommended by the reviewer, we have considered alternative approaches for comparison purposes including Random Projections (termed Random CNN in the revised manuscript) and Principal Component Based Projection (termed PCA CNN in the revised manuscript). These two methods are essentially at the two ends of the spectrum. We consider random projection because of its obvious theoretical underpinning given in Johnson-Lindenstrauss Lemma that establishes restricted isometry conditions with high probability. Furthermore, because of its computational benefit, random projections (or its sparse counterpart) has been extensive utilization in dimension reduction for image data and even dimension reduction in databases (D. Achlioptas, Database friendly random projections, Proc 20th ACM Symp Principles of Database Systems, Santa Barbara, CA, 2001, 274–281.). The choice of PCA is obvious because under Euclidean assumption, if all but two eigen values are trivial, it will strictly establish that the ambient manifold of the inputs is isometric to a Euclidean plane.

Our results indicate that REFINED CNN outperforms both Random CNN and PCA CNN for all the studies using synthetic dataset, NCI dataset and GDSC Dataset when we only consider our target dimension to be 2D. Note that due to the extreme computational complexity of nested CV, we have used nested cross validation for a random subset of cell lines to confirm that the comparisons using Nested CV and hold-out produces similar results for larger datasets. As the nested CV results in Figure 18 shows, REFINED CNN outperforms Random CNN and PCA CNN.

The following information about the alternative image generation methods are included in the section “Comparison with other image generation methods”.

“In this section, we consider two alternative image generation methods for comparison purposes: random projection -based and PCA-based. In the random projection method, we assume each image is a matrix and the location of each entry in the vector is randomly mapped to a location in the matrix. Thus, we placed each element of the drug descriptors or the gene expression on the image (matrix) coordinates one after another. Principal component analysis (PCA) [26] is mainly used for dimensionality reduction and visualization purposes, where each sample could be represented on a 2D plane aligned with their first two principal eigen vectors. The first two principal components of the transposed covariance matrix with rows

being features and columns representing samples, were selected as the feature coordinates. Some of the generated images using the random and PCA methods are shown in figure 2.”

- Study the performance as a function of the number of input features and again compare to randomly ordered 2D images

As recommended by the reviewer, we have studied the performance of the REFINED CNN as compared to other methods (including random CNN) as a function of input features with results being reported in Table 18 of the revised manuscript. We observe that REFINED CNN outperform Random CNN in terms of both NRMSE and PCC.

Minor Points:

“- It’s not clear how the random dataset was created. Which distributions were used? Were the target variables normalized as well?”

As suggested by the reviewer, the synthetic dataset section has been clarified. We used normal distribution to generate the features of the synthetic dataset. We also normalized the target values between 0-1.

“- Page 8: Normalized IC50s to the [0,1] interval. It is not clear if this was done globally or per drug, or per cell line. Also, figures 5, 6, and 7 to span the whole range. Please clarify. Why is this necessary? I am not sure this is justified.”

As recommended by the reviewer, we do not normalize IC50s in the range of [0,1] anymore to avoid misunderstanding, and all the results are provided based on the actual range in the revised version of the manuscript. As we revised the manuscript, the figure numbers are not same as before, and the table below represents which figure in the revised manuscript is equivalent to which figure in the initial manuscript.

Initial submission	Revised submission
5	17 in the appendix
6	9
7	10

In the revised manuscript, figure 17 of the appendix shows scatter plot results of the synthetic data. In figure 17, the observation values span the entire range while the prediction values of the models don’t span the same range depending on their bias. For example, since RF overpredicts lower values, and underpredicts higher values, its corresponding prediction values don’t span the entire range. On the other hand, the REFINED CNN doesn’t suffer from such a problem, therefore its values span the entire range. This happens when we deal with the GDSC data too whose results are shown in figure 10 of the revised manuscript. As shown in figure 3, the cell lines with more than 10k drug responses that are selected for the study from NCI60 data contains a unique distribution with a massive peak and thus figure 9 of the revised manuscript shows scarcity of the responses in the range below the peak.

“- Page 8: Figure 3c. The title says “IC50” but the legend says “NLOGIC50”.

- Page 16: “we used the NCI60 coordinates to generate the drug images for the 222 drugs”; it appears that this provides an unfair advantage to the REFINED algorithm over the rest of the methods.”

As suggested by the reviewer, we do not incorporate any prior knowledge anymore, and we generate completely independent coordinates using REFINED for drug data of the GDSC dataset. The results in the revised version of the manuscript are based on not incorporating any prior knowledge. Currently, we use 992 chemical descriptors for the GDSC drugs, which is larger compared to the 672 descriptors that we used for NCI60.

“- Page 10: The hybrid CNN receives inputs that are pre-organized into the two distinct data types (transcriptomics and chemical descriptors). Again, this seems to be unfair to the other algorithms. To avoid this, please also compare the performance of all algorithms based on transcriptomic data alone”

Unfortunately, the number of samples using only transcriptomics data is less than 1000 for each drug which restricts the applicability of a CNN based deep learning approach. CNN based deep learning approaches are suitable for scenarios with larger sample sizes as reported in Figure 7 of the revised manuscript where the different model performances are compared while changing the training set size. Thus, for scenarios with less than 1000 samples such as transcriptomics data alone for GDSC, it is better to apply shallow models such as RF or SVM.

“- I apologized if I missed this in the methods, but it is not clear if different models are trained per drug, or whether there is a single model (e.g. a multi-class multi-label classification). To avoid unfair comparisons, all tested algorithms should be trained to the same learning task.”

As pointed out by the reviewer, we clarified how models were trained on the same learning tasks by adding the following explanations in the predictive modeling sections.

NCI:

“We randomly picked 17 cell lines where each cell line contained more than 10k drug responses so that we have sufficient number of samples to train a deep learning model. Subsequently, 17 set of models were trained to predict NLOGGI50 of drugs tested on the selected cell line. In each set, all the above-mentioned models were trained on the same set of samples and tested on a separate set of same samples.”

GDSC:

“We trained REFINED CNN and 6 other competing models on same set of samples, where each sample is a combination of one drug tested on one cell line. All the models were tested on a separate set of same samples.”

“- Finally, I suggest that the authors come up with more creative ways to present their results in the main figures. Most of the presented results figures and tables are good and informative but are better suited for the Supplemental section (with the exception of figures 2 and 4). The main figures should visualize the information better. Also, figure 1 is very important, but needs to be improved.”

As suggested by the reviewer, we used different graphs to represents our results in more visual ways and moved majority of the tables to the appendix section.

The authors would like to thank *Reviewer No. 2* for his/her constructive review of the manuscript. With regard to the concerns raised by this Reviewer, the following changes have been incorporated in the revised version:

Major points:

“1. The presentation is fairly poor.

Above all, ‘automated feature selection’ is a misleading term. The authors emphasize automated feature selection as a core contribution of the paper, however, ‘feature selection’ seems incorrect. Instead, ‘automated feature extraction’ is an appropriate term since it is derived from CNNs that allow feature extraction.

Moreover, the authors did not elaborate specific details of how each feature value is visualized in the generated 2D image.

In addition, some fundamental notations have an incorrect definition in section 2.1.1. For example, in equation (1), q must be defined as a combination of p taken 2 because q denotes all possible pairs. Likewise, IG denotes Inverse Gaussian distribution in this paper, but it should refer to Inverse Gamma distribution which is theoretically correct. Besides, some notations are not specified. These could be considered as an incomplete understanding of the theoretical backgrounds.”

As recommended by the reviewer, we have restructured the paper and have proofread carefully while considering all the above-mentioned points. We have attempted to correct all terminology, grammatical, spelling errors. We added the following clarification in the REFINED section of the revised manuscript to explain the visualization in the generated 2D images.

“Once the cost function is minimized, a set of unique coordinates in a 2D space was produced for each feature. Using those coordinates, we mapped the features into a 2D space and create an image per sample. The created images are used to train the REFINED CNN.”

“2. In drug response tasks, three baseline models are conventional which do not offer state-of-the-art performance. As the authors imply REFINED-CNN provides better performance than others, the authors should assess the performance of REFINED-CNN comparing with other advanced models giving a state-of-the-art performance (i.e., Deep-Resp-forest, Transfer learning-based, Heterogeneous graph)”

As suggested by the reviewer we used the Deep-Resp-forest and Heterogeneous graph models for comparison, considering the fact that these algorithms were developed for different applications as explained below:

Deep-Resp-forest (DRF): DRF is a deep cascaded forest model, designed to classify the anti-cancer drug response as “sensitive” or “resistive” based on integration of heterogeneous input data (gene expression and copy number alteration (CNA)). It was applied on the GDSC and CCLE data, in a drug-wise manner, where the DRF model was trained for each drug separately. So, it can be applied on the GDSC regression task of our study considering the differences: 1) we train a regression model not a classification model, 2) inputs of our model are gene expression and drug descriptors. To address the

first difference, we can convert the random forest classifier to a random forest regressor for prediction. The second difference cannot be addressed as the second arm of the DRF is designed to handle the CNA data rather than the drug descriptor data. Therefore, the performance of DRF is poor compared to not only REFINED CNN, but also other models.

Heterogeneous graph neural network (HGNN): Majority of the HGNN studies were developed to handle text, image and other type of data for different applications than ours ranging from link prediction, recommendation and classification such as:

<https://dl.acm.org/doi/pdf/10.1145/3292500.3330961?download=true>.

Therefore, the use of these algorithms may not end up with results comparable to what we achieved with REFINED. To the best of our knowledge, the only HGNN study with drug discovery application that is close to our study is: <https://pubs.acs.org/doi/pdf/10.1021/acs.jcim.9b00387> . This study embeds the 3D structural information of protein-ligand complexes in adjacency matrix, devising a distance-aware attention algorithm to differentiate various types of intermolecular interactions, and introducing a variant of graph neural networks suitable for learning protein-ligand interaction. This approach can be used for GDSC regression task, where we use gene expression and drug descriptors for drug sensitivity prediction. We modified their method to learn gene-drug descriptor interactions using Euclidean distance matrix in the attention graph to update the node features, aggregate the node features and process features. Then we predicted the IC50 for each cell-drug combination.

The results of the DRF and HGNN as compared to REFINED CNN in terms of NRMSE, PCC and Bias are included in Table 3 of the revised manuscript and the robustness results are included in Table 16 of the Appendix and the Gap Statistics are included in Table 17 of the appendix. We observe that REFINED CNN outperforms both DRF and HGNN in terms of NRMSE, PCC and Bias and the results are statistically significant based on the robustness analysis and Gap Statistics.

“3. This paper does not clarify how components in the method contribute to the advantages REFINED gives. REFINED is composed of Bayesian MDS and hill-climbing algorithm which are global dimensional reduction and local search optimization, respectively. However, the authors do not explain specific reasons for using BMDS instead of other global dimension mapping methods such as Isomap. Also, it lacks a comparison of local search and local dimension reduction methods (e.g., Laplacian Eigenmaps, LLE). Hence, Ablation tests are desired.”

As advised by the reviewer we added an ablation study section to the revised manuscript (Section 3.5), where we used other dimensionality reduction techniques for REFINED initialization. For comparison purposes, we considered local dimensionality reduction approaches of Local Linear Embedding (LLE) and Laplacian Eigenmaps (LE) and global dimensionality reduction approach of Isomap. We also assigned feature coordinates randomly without using any dimensionality reduction technique to evaluate the hill climbing component. Detailed results and descriptions are provided in the ablation study section of the revised manuscript (section 3.5) and figures 11, 12, 23-27 and tables 4, 14 and 15.

Minor Points:

“1. In section 2.1.1, a constraint is omitted in the notation. For instance, there should be a constraint of

$j \neq k, j, k = 1, 2, \dots, p$ in estimating $x(\cdot)$ based on observed distance. Also, notation $S_{j,l}$ (in 4th paragraph, page 6) is undefined.”

As suggested by the reviewer, we have restructured the theoretical Basis for REFINED section of the manuscript and corrected all notations.

“2. Analyses are inadequate. There is no qualitative analysis that examines the effectiveness of the REFINED-CNN method. On top of that, quantitative analyses only describe results, neither interpretation nor intuition. In-depth analyses, such as case study and error analysis would increase the overall quality of the paper.”

As suggested by the reviewer, we have included the following additional analysis in the revised manuscript (a) confidence intervals of the error metrics (b) Robustness analysis based on performance on different Bootstrap test samples which illustrates that REFINED CNN predictions are robust over different test sample sets (c) Comparison of each model prediction to random predictions to show the statistical significance of the predictions. We created a Null distribution based on random predictions following the distribution of training Y and used that to evaluate how likely our prediction can be generated from a random prediction. We also added GAP statistics to compare the various models. (d) We analyzed the performance of the various models with change in sample size and showed how CNN starts outperforming other models when sample size increases (e) we analyzed the performance of our REFINED CNN approach as compared to additional models including other approaches to generate 2D images from 1D Vectors such as PCA based or Random Projections. (f) We conducted an ablation study to evaluate the significance of various steps in the methodology. We considered Local embedding approaches such as LE and LLE and global approaches using Geodesic Distance such as ISOMAP for comparison purposes with our Bayesian MDS approach (further detailed analysis of the ablation study are included in section 3.5 of the revised manuscript).

“3. Using accuracy as the only metric in classification tasks is inappropriate. It could mislead an evaluation when an output distribution is imbalanced. F1 score would be better than the single accuracy.”

As advised by the reviewer we have added F1-score, precision, recall, and AUROC along with accuracy to report the performance of each classifier.

“4. Are there any reasons to utilize a hill-climbing algorithm despite its local optima problem?”

The hill climbing is actually used because of its local optima properties. Observe that the output of the BMDS algorithm produces a set of locations \hat{s} in the unit square. This is not a veritable image. To convert this collection of points and their associated values (value of the predictor) into an image, we need to impose a grid on unit square such that at most one \hat{s}_i is contained in each grid. The hill-climbing operation is undertaken to identify the coarsest resolution of this grid. More specifically, suppose \hat{s}_i is contained in grid i and \hat{s}_j is contained in grid j . Let $\widehat{\delta}_{ij}$ be the BMDS posterior estimate of the distance between \hat{s}_i and \hat{s}_j , the hill-climbing operation is performed to minimize the distortion associated with placing \hat{s}_i and \hat{s}_j in their respective grid –centers, i.e, $\text{dist}(\text{center of grid } i, \text{center of grid } j)$.

j) $\approx \widehat{\delta}_{ij}$. Clearly, given the global optimizer producing $\widehat{\delta}_{ij}$, the hill-climbing operation identifies a set of local adjustments that closely follows the global optimum estimates

“5. Thorough details of baseline models should be provided.”

As recommended by the reviewer, we have included the details of optimized hyper-parameters of each baseline model in the “hyperparameter search” sub section of the paper. We have used multiple baseline models for comparison including Random Forest, Support Vector Machines, Artificial Neural Networks, Random Projection CNN, PCA CNN and Elastic Nets (Logistic Regression for Classification). Further details on the predictive models are included in section 2.3 Predictive Models.

“6. Figure1 does not reflect its caption. The figure illustrates GDSC dataset, whereas its caption is about NCI60 dataset.”

As pointed out by the reviewer, we have corrected the caption of figure 1.

“7. Figure 1 has a low resolution.”

As recommended by the reviewer, we have created a new figure to elaborate the REFINED diagram with high resolution.

“8. 'Voxel representation' appears to be inaccurately used with 2D image space, since voxel indicates 3-dimensional space.”

As pointed out by the reviewer, we have corrected the terminology and used pixel instead of voxel.

Extract reprinted by permission from Springer Nature: Geeleher, P., Cox, N.J. & Huang, R.S. Clinical drug response can be predicted using baseline gene expression levels and *in vitro* drug sensitivity in cell lines. *Genome Biol* **15**, R47 (2014).

Report on Modeling using Bortezomib Clinical Trial Data

As suggested by the reviewer, we tried to apply our model to patient's data (bortezomib clinical trial on multiple myeloma) and predict their response. We quote the below paragraph from [1]:

“The authors of the original clinical study reported that a 100-gene signature model [2], built on two arms of the trial (025 and 040), could predict bortezomib response in the third (039) arm of the trial with 63% accuracy. To compare our predictions with those originally reported, we assessed the performance of our model on only this third arm of the trial. The [1] models generate a continuous variable and to compare the results previously reported directly, we must dichotomize this variable (i.e. split the data into ‘sensitive’ and ‘resistant’ at an arbitrary cut-point). At the optimal cut-point (-5.29), 51 of 71 patients were correctly classified, meaning that our method achieved a classification accuracy of 72%.”

The study is interesting and informative in the paper, but we found some inconsistency while we went over their code in the supplementary. They modeled a ridge regression on *in vitro* data then predicted the log (IC50) of the patients. They mapped the predicted log (IC50) to “Responsive” and “Nonresponsive” using the patient's data information included in the test set. As it is shown in the figure 1, they iteratively tried all the possible cutoff points on the test set, to find the optimal cutoff point which maximizes the accuracy of the test set, that is not fundamentally ideal. Therefore, we decided not to include our comparison in the manuscript and provide a separate report.

Figure 1. Finding optimum cut off point in [1]

We follow same preprocessing steps as [1], including power transform, and homogenization of gene expression between two different domains. Then the preprocessed data is utilized for modeling. As the patient’s data is divided into arm A and B, we summarize our results following the same division. For comparison, figure 4-a and 4-d of [1] are provided in the appendix of this report.

A) In vivo responders and non-responders to bortezomib (compared with figure 4-a of [1]):

All analysis in [1] is done in R, while we use Python (scikitlearn and tensorflow), therefore there are some differences in the results that we have, compared to the [1]. Also, as we ran their provided R code, we noticed some difference in the results produced by the R code than the paper, which might be because of the updates on the downloaded datasets. According to the R-code, the genes with variance lower than 0.2 are discarded. Using the same variance threshold makes the genes list empty, therefore we used variance threshold as 0.1 As suggested by the reviewer we compared our REFINED CNN method with Elastic Net (EN), plus support vector regressor (SVR) and Radom Forest (RF). The results are summarized in the table 1. Figure 2 shows, accuracy of REFINED CNN versus different threshold cutoff points, where the optimum accuracy is 0.627 at log (IC50) of -4.78. The corresponding box plot is shown in the figure 3.

Table 1. Classification and regression results using different models to train on genes and predict drug sensitivity for in vivo responders and non-responders to bortezomib (REFINED CNN 2 is trained on the GDSC dataset).

Model	NMRSE	PCC	ACC	Precision	Recall	F1-score	AUC
EN	0.960	0.404	0.556	0.570	0.556	0.531	0.555
SVR	0.947	0.361	0.533	0.602	0.533	0.441	0.535
RF	0.956	0.333	0.621	0.630	0.621	0.614	0.621
Ridge	0.963	0.405	0.604	0.612	0.604	0.596	0.604
REFINED CNN	0.969	0.317	0.627	0.634	0.627	0.623	0.628
REFINED CNN 2	0.414	0.910	0.568	0.581	0.568	0.551	0.569

We used the trained model on GDSC, which incorporates the drug and cell line data as two separate inputs. Then, we used REFINED images for bortezomib, and same gene coordinates of cell line images to generate corresponding REFINED images of the patient’s data. If we use the network trained on the GDSC as reported in our paper directly to classify the patients, accuracy will be 0.568, the details are provided in table 1 as REFINED CNN 2.

On other scenario is using transfer learning to predict responders and non-responders’ patients. Hence, we replaced the last fully connected layer of our network with a fully connected layer with two nodes followed by a SoftMax layer, with a binary cross entropy loss function. Half of the patient’s data were used for fine tuning, and other half for testing. The results of fine-tuned transferred network improved the accuracy, precision, recall, F1-score, and AUC up to 0.651, 0.672, 0.651, 0.629, 0.6347.

Figure 2. Accuracy versus different threshold cutoff points for predicted $\log(\text{IC}_{50})$ by REFINED CNN on arm A of the dataset.

Figure 3. boxplot of predicted $\log(\text{IC}_{50})$ values by REFINED CNN of arm A of patient dataset separated

B) Arm (039) of bortezomib trial (compared with figure 4-d of [1]) :

The same preprocessing as Arm A was done on the training data and tested on the arm B of the dataset. The results are summarized in table 2. The same transfer learning approach as part A, were used for predicting responder and non-responder patients of arm (039) of the bortezomib trial. The results of fine-tuned transferred network improved the accuracy, precision, recall, F1-score, and AUC up to 0.764, 0.779, 0.764, 0.791, 0.735.

Table 2. classification and regression results predicted by different algorithms for the arm (039) of the bortezomib trial (REFINED CNN 2 is trained on the GDSC dataset).

Model	NMRSE	PCC	ACC	Precision	Recall	F1-score	AUC
EN	0.960	0.404	0.634	0.635	0.634	0.626	0.624
SVR	0.987	0.396	0.634	0.638	0.634	0.622	0.622
RF	0.929	0.378	0.620	0.641	0.620	0.589	0.601
Ridge	0.960	0.404	0.606	0.618	0.606	0.577	0.588
REFINED CNN	1.026	0.301	0.676	0.688	0.676	0.675	0.681
REFINED CNN 2	0.414	0.910	0.620	0.627	0.620	0.601	0.601

Figure 4. Accuracy versus different cut off threshold for predicted log (IC50) by REFINED CNN on arm (039) of bortezomib trial.

Figure 5. Boxplot of predicted log (IC50) values by REFINED of arm (039) of bortezomib trial.

Figure 6. classification accuracy versus different cut off thresholds from R-code of supplementary information of [1]

Conclusion

Our experiment shows that finding the optimum cut off point to maximize accuracy of predicting non-responder or responder patients to Bortezomib, is completely independent from minimizing error of

regression model trained on in vitro data. In other words, since we use the test data information to optimally predict the test data, therefore optimizing any regression model on the in vivo data is not bested. According to the tables 1 and 2, the models with minimum NRMSE doesn't necessarily provide better classification accuracy. Also, as it shown in the figure 6, the optimum classification results are acquired by picking an unstable cut off threshold, where if the cut off point changes with a small degree, then the classification results will be decreased significantly.

In these scenarios' authors recommend using deep or statistical transfer learning approaches to predict responsive and non-responsive subjects.

- [1] P. Geeleher, N. J. Cox, and R. S. Huang, "Clinical drug response can be predicted using baseline gene expression levels and in vitro drug sensitivity in cell lines," *Genome biology*, vol. 15, no. 3, p. R47, 2014.
- [2] G. Wright, B. Tan, A. Rosenwald, E. H. Hurt, A. Wiestner, and L. M. Staudt, "A gene expression-based method to diagnose clinically distinct subgroups of diffuse large B cell lymphoma," *Proceedings of the National Academy of Sciences*, vol. 100, no. 17, pp. 9991-9996, 2003.

APPENDIX

Figure 7. Figure 4-1 of [1]

Figure 8. Figure 4-d of [1]

Figure 4a & d reprinted by permission from Springer Nature: Geeleher, P., Cox, N.J. & Huang, R.S. Clinical drug response can be predicted using baseline gene expression levels and *in vitro* drug sensitivity in cell lines. *Genome Biol* **15**, R47 (2014).

Reviewer #1 (Remarks to the Author):

The authors have improved their manuscript and have addressed all my comments.

Reviewer #3 (Remarks to the Author):

REFINED (Representation of Features as Images with Neighborhood Dependencies): a novel feature representation for Convolutional Neural Networks

The strategy of converting a feature similarity space into an image is novel. The authors revised the manuscripts according to the comments of the previous round. However, the manuscript still suffers from a significant amount of ambiguities on the methods and the results. Please see my detailed comments below:

Major comments:

1. Figure 1: the example data at STAGE1 seems displaying cancer cell line features (e.g. gene expression) only. How to incorporate the molecular descriptors of the drugs into the table? The PaDEL descriptors are for the drugs, not for the cell lines, but in the example data I don't see any of the PaDEL descriptors (usually they are binary or count data). In the STAGE2, the distance matrix contains negative values, which might be difficult to interpret. In the main text (section 2.1) it was mentioned that the distance is Euclidean, so that cannot be negative?
2. The idea of MDS (and its Bayesian version) was to map the features into a 2-D space, for which the coordinates are no longer bound to the size of the feature vector. After the REFINED stage, the coordinates of the features were adjusted to be fit into a square matrix, but then the actual distances between the features were lost. For example, F1-F14 and F1-F10 became equally distanced as both of them are one-degree neighbors, despite that their actual distances might differ. With such a loss of information, it becomes less clear about how the CNN can compensate for it?
3. The method: on page 4 the parameter d_{jk} is defined as 'observed distance between the i th and j th predictor', while the parameter δ_{jk} is defined as 'true but unobserved distance'. Is d_{jk} the distance on the 2D map while δ_{jk} is the distance based on the high-dimensional data?
4. Convergence of the algorithm: how good the hill climbing algorithm converges to the global optimal? How the performance of the algorithm is affected if the algorithm does not reach the global optimal but local optimal?
5. Figure 2 showed the examples of REFINED images for different drugs. What is the NSC ID? Is it a drug ID? What is the color code? This information should be described in the figure legend.
6. Figure 2: it seems that the PCA-based feature map has more dark areas (null values) compared to Random and REFINED maps. Should they have the same number of null values? i.e. these different mapping methods change only how the feature vectors are distributed spatially.
7. Random vs PCA: it seems that these two feature maps provide similar performances (e.g Figure 5 and Table 10), which is of concern. Does it imply that adding feature similarity information does not necessary outperform the 'uninformative' feature map? I understood that only one random feature map was generated for all the comparative analyses. If you generate multiple random feature maps would you confirm that the REFINED map is significantly better?
8. Table 2: the best performance of bias is achieved at PCA for the scenario of training on 80%, not at REFINED.
9. NCI data analysis. What is the unit of the GI50? micromolar or nanomolar? In Figure 3, the number of samples corresponds to the number of drugs?
10. The NCI60_GI50_normalized_small file. There is a 'NORMLOG50' column in the file, which was not reported in the main text. Is it the same as NLOGGI50?
11. The threshold of NLOGGI50. It was mentioned that 4.5 was used as the threshold on page 9, while on page 14 it was reported as 4.25.
12. GDSC data analysis. What is the advantage of generating images separately for a drug and for a cell line, as compared to have one image for a drug-cell line pair?

Minor comments:

1. Page 7: 'one illustrative cell lines is shown in figures 3' -> 'Figure 3'.
2. Figure 4: x label should be 'Sample size'.

Detailed Response to Reviewer Comments (“REFINED (REpresentation of Features as Images with NEighborhood Dependencies): A novel feature representation for Convolutional Neural Networks”)

REVIEWER COMMENTS

Reviewer #1

“The authors have improved their manuscript and have addressed all my comments.”

The authors would like to thank Reviewer No. 1 for appreciating the changes made in the revised manuscript.

Reviewer #3

The authors would like to thank Reviewer No. 3 for his/her constructive review of the manuscript and his/her valuable comments. With regard to the concerns raised by the Reviewer, the following changes have been incorporated in the revised version:

Major points:

“1. Figure 1: the example data at STAGE1 seems displaying cancer cell line features (e.g. gene expression) only. How to incorporate the molecular descriptors of the drugs into the table? The PaDEL descriptors are for the drugs, not for the cell lines, but in the example data I don’t see any of the PaDEL descriptors (usually they are binary or count data). In the STAGE2, the distance matrix contains negative values, which might be difficult to interpret. In the main text (section 2.1) it was mentioned that the distance is Euclidean, so that cannot be negative?”

The earlier Figure 1 image contained images of general data matrices without being specific to PaDEL descriptors and Euclidean Distances. Based on reviewer recommendation, we have altered the figure to incorporate matrices representing PaDEL descriptors and Euclidean distances. Furthermore, since the overall description in the figure is for the NCI60 scenario with multiple drugs and each cell line, we have modified the figure caption according to the changes made in the figure.

“2. The idea of MDS (and its Bayesian version) was to map the features into a 2-D space, for which the coordinates are no longer bound to the size of the feature vector. After the REFINED stage, the coordinates of the features were adjusted to be fit into a square matrix, but then the actual distances between the features were lost. For example, F1-F14 and F1-F10 became equally distanced as both of them are one-degree neighbors, despite that their actual distances might differ. With such a loss of information, it becomes less clear about how the CNN can compensate for it?”

Adjustment of the coordinates from unbounded MDS plane to a compact support does not lead to any loss of information per se. The relative ordering of the distances is preserved. To motivate the projection on unit square, one can envision working with scaled version of the original distance ($d_{ij}/\max(d_{ij})$). The projections are only giving a configuration of the location of the predictors and once the locations are estimated, the actual values of the predictors are plugged into that pixel. So, there is no loss of information in terms of actual predictor values. Furthermore, it is customary in stringing literature (using UDS instead of MDS) to impose a [0,1] support for latent stochastic process (see Chen et al.,[32])

However, we note that loss of information does occur when more than 1 predictor occupy the same pixel as it happens in the PCA procedure. Consequently, the image appears to be spuriously sparse and

leads to inferior predictive performance. The key, therefore, is the latent theoretical formulation and the hill-climbing post-processing that enforces at most 1 predictor per pixel.

“3. The method: on page 4 the parameter d_{jk} is defined as ‘observed distance between the i th and j th predictor’, while the parameter δ_{jk} is defined as ‘true but unobserved distance’. Is d_{jk} the distance on the 2D map while δ_{jk} is the distance based on the high-dimensional data?”

d_{jk} is the observed dissimilarity in the high-dimensional data. δ_{jk} is the distance in 2D plane. We precisely define δ in pp 4 (last paragraph) of the manuscript. Observe that δ is a function of unknown spatial locations (s_j) and hence capture the distance among predictors in 2D plane.

“4. Convergence of the algorithm: how good the hill climbing algorithm converges to the global optimal? How the performance of the algorithm is affected if the algorithm does not reach the global optimal but local optimal?”

We have described in detail the relevance of hill-climbing algorithm in response to reviewer 1’s comment. First, we note that hill-climbing operation, in this methodology, is purely a local adjustment. The global optimum is produced by the BMDS solution of δ . To reiterate, once the locations (s_i) are obtained from BMDS algorithm and a square tessellation is imposed, s_i can occupy any location in the grid that it is assigned to. Suppose \hat{s}_i is contained in grid i and \hat{s}_j is contained in grid j . Let $\widehat{\delta}_{ij}$ be the BMDS posterior estimate of the distance between \hat{s}_i and \hat{s}_j , the hill-climbing operation is performed to minimize the distortion associated with placing \hat{s}_i and \hat{s}_j in their respective grid-centers, i.e. $\text{dist}(\text{center of grid } i, \text{center of grid } j) \approx \widehat{\delta}_{ij}$. Clearly, given the global optimizer producing $\widehat{\delta}_{ij}$, the hill-climbing operation identifies a set of local adjustments that closely follows the global optimum estimates. Since hill climbing has an explicit goal to minimize the distortion of $\widehat{\delta}_{ij}$ subject to only local perturbations of \hat{s}_i , convergence is achieved fast. In fact, observe that the BMDS actually produces posterior distributions of $(\delta_{jk} \forall j < k = 1, 2, \dots, p)$. So, tolerance of the hill-climbing algorithm can be explicitly stated by imposing the hill-climbing related distortion to be contained within coordinate specific HPD interval of δ .

“5. Figure 2 showed the examples of REFINED images for different drugs. What is the NSC ID? Is it a drug ID? What is the color code? This information should be described in the figure legend.”

We appreciate reviewer’s suggestion for clarification of NSC ID. We have modified the figure caption to describe NSC ID based on the definition of NSC ID/number that can be found on NCI website . We have also included a colorbar for clarification.

“*NSC Number: Originally known as Cancer Chemotherapy National Service Center number, the NSC number is an identifying number assigned by DTP to an agent or product (e.g. small molecule or biological agent). Most structures have a unique NSC number but over the years a small percentage of structures or agents may have been assigned more than one NSC number. In the case of salts, different salt forms (e.g. HCl, HOAc) are designated with separate NSC numbers.”

“6. Figure 2: it seems that the PCA-based feature map has more dark areas (null values) compared to Random and REFINED maps. Should they have the same number of null values? i.e. these different mapping methods change only how the feature vectors are distributed spatially.”

*We appreciate the reviewer comments on this perceived anomaly. Note that the mapping for our REFINED approach consists of two steps (i) a dimensionality reduction (such as Multi-Dimensional Scaling) followed by (ii) a search optimization algorithm that ensures at most one feature occupy a single pixel. If we do not use the second step of search optimization algorithm (hill climbing), many features can occupy a single unconstrained pixel. This happens because, dimensionality reduction approaches such as PCA or MDS produces co-ordinate locations that are real numbers which when converted to integers in an unconstrained fashion (as images consists of discrete pixels) multiple real coordinates get assigned to the same integer pixel. Since the PCA-based feature map shown in figure 2 does not include the second optimization step, a number of features are overlapping and thus the 672 features for the PCA case are represented by significantly less number of pixels in the 26*26 image (676 pixels) and the remaining pixels not containing any features are shown as black (value 0). Consequently, this image is spuriously sparse and leads to loss of information as multiple features assigned to the same pixel are no longer identifiable and the pixel intensity is the average values of these features.*

*On the other hand, REFINED incorporates the second step that ensures that no features overlap in the discrete pixel space and thus the number of zero values in the 26*26 image (= 676 pixel image) for 672 features is only 4. We had also set up the random projection constraint to randomly create a coordinate for each feature without any overlap with other features and thus the zero values for the random map is also 4.*

Thus, the number of zero (null) values associated with PCA projection is more than REFINED or Random maps.

“7. Random vs PCA: it seems that these two feature maps provide similar performances (e.g Figure 5 and Table 10), which is of concern. Does it imply that adding feature similarity information does not necessary outperform the ‘uninformative’ feature map? I understood that only one random feature map was generated for all the comparative analyses. If you generate multiple random feature maps would you confirm that the REFINED map is significantly better?”

Note that our ablation study in section 3.5 shows that both the steps of (a) dimensionality reduction and (b) the search optimization to remove overlap are important for improving the predictive performance. Since the PCA map does not include the second step of search optimization (that ensures features do not overlap), the performance of PCA map is expected to be lower. The random map has been setup in a way that ensures no overlap of features and thus the performance is similar to PCA.

As suggested by the reviewer, we have created multiple random projections to explore variations with different random maps and provided the results for the multiple random projections in table 11 (appendix) of the revised manuscript. None of the multiple random map projection performance exceeded the performance of REFINED CNN. The REFINED CNN performance exceeded the average performance of the multiple random projections by more than 10% for NRMSE, PCC and Bias.

“8. Table 2: the best performance of bias is achieved at PCA for the scenario of training on 80%, not at REFINED.”

Thank you for pointing that out, we have corrected the typo.

“9. NCI data analysis. What is the unit of the GI50? micromolar or nanomolar? In Figure 3, the number of samples corresponds to the number of drugs? “

As requested by the reviewer, we have included the unit of GI50 (in molar, for instance $N\log GI50 = 8$ represents $GI50 = 10^{-8}$ molars = 10 nanomolar) in NCI Data analysis. In figure 3 (a & b), the number of samples is equal to number of drugs applied on the selected cell line of the NCI60 dataset (Note that not all drugs were tested on all cell lines). In figure 3 (c), the number of samples is equal to all the drug-cell line combinations in GDSC dataset.

“10. The NCI60_GI50_normalized_small file. There is a ‘NORMLOG50’ column in the file, which was not reported in the main text. Is it the same as NLOGGI50?”

As requested by the reviewer, we have explained ‘NORMLOG50’ in the main text. NORMLOG50 refers to the normalized NLOGGI50.

“11. The threshold of NLOGGI50. It was mentioned that 4.5 was used as the threshold on page 9, while on page 14 it was reported as 4.25.”

Thanks for pointing out this typo, we corrected all the thresholds in the manuscript to 4.25.

“12. GDSC data analysis. What is the advantage of generating images separately for a drug and for a cell line, as compared to have one image for a drug-cell line pair?”

Thanks for pointing out this question. The domains of drug descriptors (describing the chemical properties of drugs) and gene expressions for cell lines (providing a measure of how various genes in the cell line have been expressed) are completely different. One can simply append all the features, compute the dissimilarity matrix and then create a single REFINED image. But there is no reasonable way to interpret the dissimilarity between a molecular descriptor and a gene and hence the dissimilarity matrix is neither interpretable nor justifiable. Thus, we generated separate images for each drug and cell line such that the convolutional layers of CNN can extract features from each image automatically and then append the extracted features in the dense layer to calculate the drug sensitivity prediction. Note that a single image for a drug cell line pair would have been possible if we had gene expressions after drug application information or if we had the detailed information on the drug targets which could have been used for estimating the gene expression following drug application- essentially, computing the dissimilarity associated with drug-gene interaction.

“Minor comments:

1. Page 7: ‘one illustrative cell lines is shown in figures 3’ -> ‘Figure 3’.

We have corrected the typo in the revised manuscript.

2. Figure 4: x label should be ‘Sample size’.”

We have corrected the typo in the revised manuscript.